# Rural Agrotourism Development Strategies in Less Favored Areas: The Case of Hacienda Guachinango de Trinidad

Norberto Pelegrín Entenza [1,2] , Antonio Vázquez Pérez [3,*] and Analién Pelegrín Naranjo [1]

[1] Doctoral Programme in Tourism, University of Alicante, San Vicente de Raspeig, 03690 Alicante, Spain; norberto.pelegrin@utm.edu.ec (N.P.E.); apn28@alu.ua.es (A.P.N.)
[2] Faculty of Administrative and Economic Sciences, Tourism Career, Technical University of Manabi, AVENIDA URBINA, Portoviejo 130105, Ecuador
[3] Faculty of Mathematical, Physical and Chemical Sciences, Technical University of Manabi, AVENIDA URBINA, Portoviejo 130105, Ecuador
[*] Correspondence: antonio.vazquez@utm.edu.ec; Tel.: +593-995858488

**Abstract:** The objective of the work was to assess and identify potential perspectives for the development of agrotourism activity in the Hacienda Guachinango del Valle de los Ingenios in the Trinidad municipality, so that the methodology applied and the results achieved can serve as a reference for application in other similar contexts in the country and in the Central American area. For this, the analysis and synthesis of spatial elements, discussion groups, the interpretation of descriptive statistics, and the study of the natural and social environment were applied. The diagnosis of resources was carried out in this work to determine the degree of competitiveness and attraction of the agrotourism product, and it was possible to establish the most convenient strategies to follow, considering the real possibilities of prospective development of the place studied. It was concluded that the proposal to develop agrotourism in Hacienda Guachinango constitutes a viable option to design and introduce a strategy for growth and socioeconomic reactivation based on the use of the natural, historical, cultural, and social potentialities of the small families dedicated to agricultural work.

**Keywords:** agricultural reactivation; agrotourism; rural tourism

## 1. Introduction

At present, on the world scale, there are extensive agricultural areas that suffer from socioeconomic depression. They have a high level of marginality characterized by precarious and poorly paid agricultural production systems. Hence, there is a need to seek alternative socioeconomic solutions that allow for an increase in income and income from non-agricultural activities, within which agrotourism stands out.

By the end of the 20th century, studies by [1,2] estimated that 51% of accommodation in Austria was farm-based and there were 22,000 farm-based sites in France. It is estimated that 7.5% of all Austrian farmers offer tourist accommodation [3].

Agrotourism is a relatively new concept that in recent years has been gaining popularity among tourists. In Spain, 1,888,639 Spanish residents and 181,998 foreign visitors decided to stay in rural tourist establishments, despite the crisis generated by COVID-19 [4]. Agrotourism can become an economic solution for the rural family, serving as an engine for sustainable development, improving social conditions and favoring the reduction of migration to the cities, especially in the current scenario that is characterized by a health emergency and the necessary process of adaptation to climate change.

Agrotourism offers multiple economic and non-economic benefits [5,6]. In reference to the economic aspect, an increase in agricultural income is sought, while at the same time it becomes a direct support for agricultural activity. Some authors such as [5,7–10] state that the diversification of company portfolios on farms dedicated to agritourism increases

income and alleviates the economic difficulties of families. They point out that especially in the United States of America, agrotourism has increased in recent years.

Agrotourism can generate environmental and sociocultural benefits. In an evaluation of the sustainability of farms, superior results were quantified in comparison to other agricultural facilities. Evaluation of the data collected from 873 farms in the United States of America with a diversified business portfolio showed that agritourism farms are more sustainable than their counterparts in terms of environmental, sociocultural, and economic benefits for farm labour, families, and society [5].

Busby and Rendle [7] point out that the traditional approach to agrotourism has evolved in recent years from a complementary commercial activity to a tourism mode in its own right, in line with authors such as [11–13], who predict further growth in demand for this tourism product. It is emphasized that from a psychological point of view, sharing the experiences of life in a country house with accommodation, breakfast, and family-style hospitality is a different attraction for the tourist [14].

There are scholars who put forward other criteria related to the agritourism modality and point out that further theoretical research is required to assess the extent to which this is a valid assumption. It is pointed out that tourism can be developed to such an extent that the profits from the activity exceed those from agriculture, which in practice leads to a gradual abandonment of agricultural work. Ref. [15] point outs that agricultural activity cannot be subordinated to tourism, as it loses its productive essence and with it the productivity of food.

The above discussion allows us to see the relevance of the theoretical study of the concept and practical criteria of the agrotourism model, which is corroborated in the ideas of [16], which points out in its studies that agrotourism still lacks an integral body of knowledge and an adequate theoretical framework due to problems with the definition. Ref. [17] points out that the term is sometimes used interchangeably with rural tourism. Ref. [18] points out that there are not enough studies related to the growth and development of agrotourism and the need to understand the dynamics of such business ventures. The demand of tourists who prefer the agrotourism modality generates new jobs within the agricultural family, especially for women, who can become direct beneficiaries of economic benefits [6,8,19,20]. The promotion of family unity to obtain collateral economic returns resizes family ties in rural communities [8,20–22] and allows the preservation of family agricultural heritage by passing the business from one generation of farmers to the next. Agrotourism manages to promote the conservation of intangible heritage, represented in local customs, ancestral practices, and rural landscapes [23–26], In practice, this becomes permanent work objectives on which the stability of the agrotourism enterprise depends. All the above stimulates local economies [27].

Agrotourism as a modality of rural tourism emerged in Europe and quickly spread throughout Latin America as a strategy for the use of natural resources by valuing agricultural activity to obtain collateral economic income for the rural family, which favors reducing the migration of the countryside to the city and achieving more stability in agricultural production.

Some authors such as [28–30] outlined their research in the study of strategies aimed at taking advantage of the recreational potential of agricultural systems in integrated rural development based on endogenous and sustainable practices through the analysis of the situation and perspectives of agrotourism activity in the upper and easternmost part of the Granada region of Las Alpujarras. In the results of their work, some strategies can be seen to take advantage of the agrotourism potential of the agrarian systems of the area. It is recommended to analyze the offer and its possible segments; design a program at the local level for the different segments; adjust the offer to the preferences of the demand; promote, advertise, and use logos; promote training through courses for farmers, technical and business training programs, study practices, and trips to appreciate the experiences in other similar projects in other Spanish territories; prioritize the observation of the maintenance of traditional Mediterranean polycultures; and carry out reconversion of

agrotourism considering the future and the improvement of access and communication between municipalities.

Other researchers such as [31,32] propose the sustainable management of self-sustaining agro-ecotourism farms as a rural development strategy for the Cevallos canton in the province of Tungurahua, Ecuador. The proposal establishes a production model for the proper management of ecotourism farms and includes zoning in its design, in which eight zones, trails, signage, areas, services, care, and recreation are determined, as well as the parking lot area. The social, economic, and ecological benefits are also determined. Practical measures for farm management in the interest of increasing production are included. The social, economic, and ecological benefits are also determined in order to satisfy the food needs of the family and to use the surplus for economic marketing. The income is used for health, education, clothing, housing, recreation, and to raise the quality of life and the socioeconomic level of the farmer by strengthening the farm with a view to the future of the family.

Ecological management preserves the natural balance of the soil, maintains soil fertility, reduces erosion, and preserves biological populations. Crops are healthier and consumers eat healthier food. Social benefits include job stability, individual and family well-being, and self-esteem in the context of increased social and community participation; the reduction of poverty and marginalization; the preservation of ancestral and cultural values; the strengthening of human values; the promotion of continuous improvement and creativity of the farmer; and the flourishing of handicraft activities developed within the community framework [31].

Practical measures for farm management are included in the interest of increasing production, emphasizing the introduction of innovations that are low-cost for the farmer, such as the implementation of organic manure, seed selection, and organic fertilization, in order to increase yields and apply other low-cost technologies [32]. In the San Carlos parish of the Naranjal canton in Ecuador, an agrotourism proposal was developed, the purpose of which is the economic and social development of the area located in a rural territory. Problems and causes that prevent its development are identified, which can affect the development of tourism alongside agriculture and sustainability [33]. Among the problems is the lack of knowledge on the part of the population about agrotourism and the absence of places to carry out this activity adequately. The causes that limit the management of the activity may be the lack of preparation related to tourism activities, the scarcity of tourism development in the area, and the lack of tourism promotion.

The author of [34] proposes a pilot system as an alternative to contribute to the development of agrotourism. The importance of the sustainable promotion of the site selected as an object of study in the Montalvo Canton of the province of Los Ríos in Ecuador was analyzed. In this work, the profile of the visitor related to sustainable agrotourism activities was obtained. It was suggested to enable an agrotourism route that integrates the natural, cultural, and productive resources of the site. The proposal included the map of the route, the offer of services, and the tour package.

In Cuba, several investigations into agrotourism have been carried out, in which various analyses and proposals are reflected upon from different angles [35–37]. As a background to the research, it can be said that since 1990, the Cuban state has deployed a political will to strongly develop the tourism sector in the country. In particular, the central region of Cuba has several historical sites typical of colonial cities, and its geographical enclave between the mountains and the coastline is an outstanding destination that, together with sun and beach, nature, and cultural tourism, can be used for tourism development [38,39]. In this context, there are state and privately owned farms, agricultural enterprises, and plantations with sufficient tourist attractions that are not being exploited and that could provide economic and social benefits for the people living in rural areas [40,41].

All the works agree that agrotourism constitutes a modality of rural tourism, which aims to achieve the participation of the peasant families in the search for collateral economic benefits. For this, agriculture is integrated as a key piece in the global strategy for sustain-

able development and the use of resources and natural attractions from agro-ecological practices that seek economic growth and environmental protection. Agrotourism stands out as a local initiative aimed at family development and rural communities in a process aimed at transforming, integrating, and strengthening socioeconomic relations between agricultural and non-agricultural activities.

Even though agricultural production is one of the main supports of the Cuban economy and the optimal use of its resources allows a close link to be established with tourism, peasants and state actors have not been able to internalize the possible economic impacts and sociocultural, environmental, and long-term governance of agrotourism, which has led to the underestimation of this type of tourism in the country.

Taking as a scenario the Hacienda Guachinango, which is located in the Valle de los Ingenios de Trinidad in the central region of Cuba and which has been declared a Cultural Heritage of Humanity site by the United Nations Organization for Education, Science and Culture, the present study focuses on the relationships that are established between the different agricultural activities that are developed, their link with tourism, and the impact of agrotourism on the production system and the local agricultural population, from the reactivation of socioeconomic activities in the field of agriculture.

Theoretically, the study is justified by the need to provide aspects related to the agrotourism modality and contrast the ways in which it is presented. At the same time, elements that contribute to the development of other investigations that venture into the said line of investigation may be offered, which justifies the methodological value. Its practical utility lies in the fact that, based on the results obtained, methods can be established that allow the results of the management of agrarian policies to be improved for less favored areas in relation to tourist activity.

The Organisation for Economic Co-operation and Development defines the most disadvantaged areas as those where social, economic, and environmental problems are concentrated. In Cuba's rural areas, as in other Latin American countries, the most economically, socially, and environmentally disadvantaged areas are located. The insufficient availability of services associated with water, electricity, and sanitation coverage are accumulated problems from previous periods. The lowest salaries are in the rural sector, and the precariousness of agricultural work means that rural areas accumulate the greatest poverty [42]. This corresponds to what has been pointed out by the authors of [43], who offer an analysis of the economic and social situation in Cuba. The deterioration of the physical and constructive structure of housing and the poor state of communication routes to rural communities, together with the reduced transport coverage for the mobility of the population [44–46], create a real situation of isolation and marginalization that worsens the living conditions of rural Cuba.

The World Tourism Organisation's recommendations on tourism and rural development state that the equitable distribution of the benefits of tourism can enhance job creation, protect natural resources and cultural heritage, promote social inclusion, and empower local communities and traditionally disadvantaged groups such as women, youth, and indigenous peoples. Inclusive tourism can contribute to making rural territories more accessible to people and visitors from different generations and with different access needs, thus creating a better quality of life for all [47].

The objective of this study was to assess and identify potential perspectives for the development of agrotourism activity in the Hacienda Guachinango del Valle de los Ingenios in the municipality of Trinidad, so that the applied methodology and the results achieved can serve as a reference for application in other similar contexts in the country and in the Latin American area.

*Theoretical Delimitation*

In recent years, agricultural activity has experienced a decrease and a notable reduction in investment. This is due to several aspects that occur in rural areas, such as the lack of adequate jobs, low wages, the precariousness of work in the agricultural sphere, the poor

infrastructure for the provision of basic services such as electricity, healthy water supply for irrigating crops, low access to technological communication services, and inadequate and in some cases no public aid [48–50].

It can be seen that these aspects have an important influence, making rural life difficult and in many cases precarious. This results in the progressive migration of people from the countryside to the cities [51–53]. The analysis of the situation exposed above serves as motivation to adopt a redesign of agricultural development policies, which advocate the multifunctionality of the territories [54,55], and in this context, rural tourism can provide practical answers to reduce the effects of the problem raised [56,57].

It is necessary to evolve from simple policies based on food production to others in which new functions are linked, such as the use of landscape values and the enhancement of native resources [58,59]. A new mode of organization is required that starts from the local scale as a driving force of the development of rural tourism [60–62] and as a dynamic axis of socioeconomic development in rural areas, expressed in the generation of jobs and income, in the care and improvement of natural spaces, and in the attraction of investment processes [63,64].

Some authors [65,66] note that rural tourism is a productive form that is based on the ability to create values by taking advantage of the natural conditions in rural territories. As a tourist activity, it allows nature, culture, gastronomy, routes, and accommodation in rural areas to be shown.

The World Labor Organization defines rural tourism as the set of activities that take place in a natural environment, the premise of which is contact with nature, local society, and its traditions. Within these modalities, adventure tourism, ecological tourism, agrotourism, ranch tourism, and cultural rural tourism among other initiatives can be observed [67,68].

Rural tourism supposes a complementary income to agricultural activity, which translates into an increase in income and an improvement in the quality of life of the inhabitants in rural areas [69]. To this end, it promotes service agriculture, through which, along with production, transformation, and marketing of agricultural products, certain tourist services are provided, along with the preservation of cultural and heritage values. This favors a multipurpose solution that satisfies the demand of rural tourism and at the same time benefits agricultural production [70,71].

Some authors such as [72,73] specify that the rural tourism model as a generalizing concept includes agrotourism or tourism in the farmer's home, like any other differentiated tourism activity that takes place in rural areas.

In an investigation carried out in the Camajuaní municipality located in the central part of Cuba, it is concluded that the rural areas of the territory represent alternatives for economic revival from the development of agrotourism, which is necessary for the economic and social development of the rural family through the different tourist products that characterize this modality [74]; however, the studies carried out showed that the tourist products associated with agrotourism are not being used properly.

Agrotourism has been mistakenly interpreted as rural tourism when it constitutes a subset of it [75]. It can be defined as a type of rural tourism that is integrated from a group of activities related to farming.

This coincides with [76], which expresses that "the concept of agrotourism does not have a clear and unanimous definition, it depends on the country or even on the different regions within it" (p. 9).

Tourists who prefer agritourism tend to come into direct contact with and enjoy the traditions and the rural family way of life, the traditional productive customs of the house in which they stay, and the transformation processes as well as the culinary customs characteristic of the environment in which they are found.

Agrotourism can be understood as a form of rural tourism, since it takes place in said environments, fields, mountains, and small towns in which the landscapes, nature, and the environment as a whole can be enjoyed along with the gastronomy and the customs.

Among the investigations that have deepened the analysis of the agrotourism modality, we can mention [77,78], which made contributions related to the revitalization and revaluation of agribusiness and rural agricultural activity. Another study [79] investigated agrotourism from the perspective of the association in Chile, which characterized the creation of agrotourism circuits based on experiences in Chiapas, Mexico.

Authors such as [80] propose an inclusive business model based on agrotourism to boost the family economy in Chimborazo, Ecuador. Previously, [81] analyzed the existing articulations between the agrarian sector, the rural spaces, and the environment at the national level.

In the same sense, [82] proposes the design of an agrotourism product as an axis to enhance local development, based on the La Esperanza site in the Bolívar canton in Ecuador.

The previous definitions do not contradict each other but rather are complementary; however, some due to their general nature and others particular to a certain area do not adjust to the objective proposed in the work, which is understood to start from the critical analysis of the contributions of [83–86], which propose the following conceptual definition of agrotourism: any tourism and leisure activity linked to any agricultural, livestock, fishing, and/or agricultural activity, with accommodation in the farmhouses and tasting products of the field to learn about traditional practices of cultivation, harvesting, and processing of agricultural, forestry, and fishing products, as well as local crafts and culture.

The design of the research starts from the operational definition of agrotourism referred to above and from it the realization of the basic studies to carry out a diagnosis of the scenario, which would allow agricultural resources as well as the inventory of the existing tourist facilities in the area to be determined.

## 2. Materials and Methods

### 2.1. Study Area

Hacienda Guachinango is in the central west region of Valle de los Ingenios, declared a World Heritage Site by UNESCO in 1988 and belonging to the municipality of Trinidad in the province of Sancti Spíritus, Cuba. It is surrounded by an agricultural area of 190 hectares of mostly typical tropical brown soils, with real possibilities to develop the traditional cultivation of sugarcane. It is characterized by tropical grasslands suitable for pastures and, to a lesser extent, typical alluvial soils, on which vegetables and fruit trees are grown. The Ay River maintains the water balance necessary to sustain the diversity of agricultural production throughout the year. Figure 1 shows the location of Hacienda Guachinango.

Hacienda Guachinango was built on a small hill 15.8 m high, and its heritage value is a given as it is the only exponent of a linked dwelling with the rural architecture of the first half of the 19th century in the Valle de los Ingenios. It is a small agrotourist agricultural enterprise, where an old colonial structure that served as the house of the landowners is located. The main agricultural activity is the cultivation and harvesting of sugarcane, small fruits, and occasional short-cycle crops for family consumption. Small and large livestock are raised to satisfy the needs of the family and to market the surplus.

In the vicinity of the Guachinango Hacienda, there is a tourist centere located in the rural village of Manaca Iznaga and two other agricultural estates with similar characteristics, where agrotourism ventures can be carried out.

There is a tourist train that runs through the Valle de los Ingenios and reaches the Guachinango Farm. This provides a reliable source of national and international tourists.

Located at approximately 16 km from the city of Trinidad, the farm can be easily reached by bus or light vehicle through the highway from Trinidad to Manaca Iznaga and Condado, as well as by taking the tourist train that runs through the Valley of the Wits.

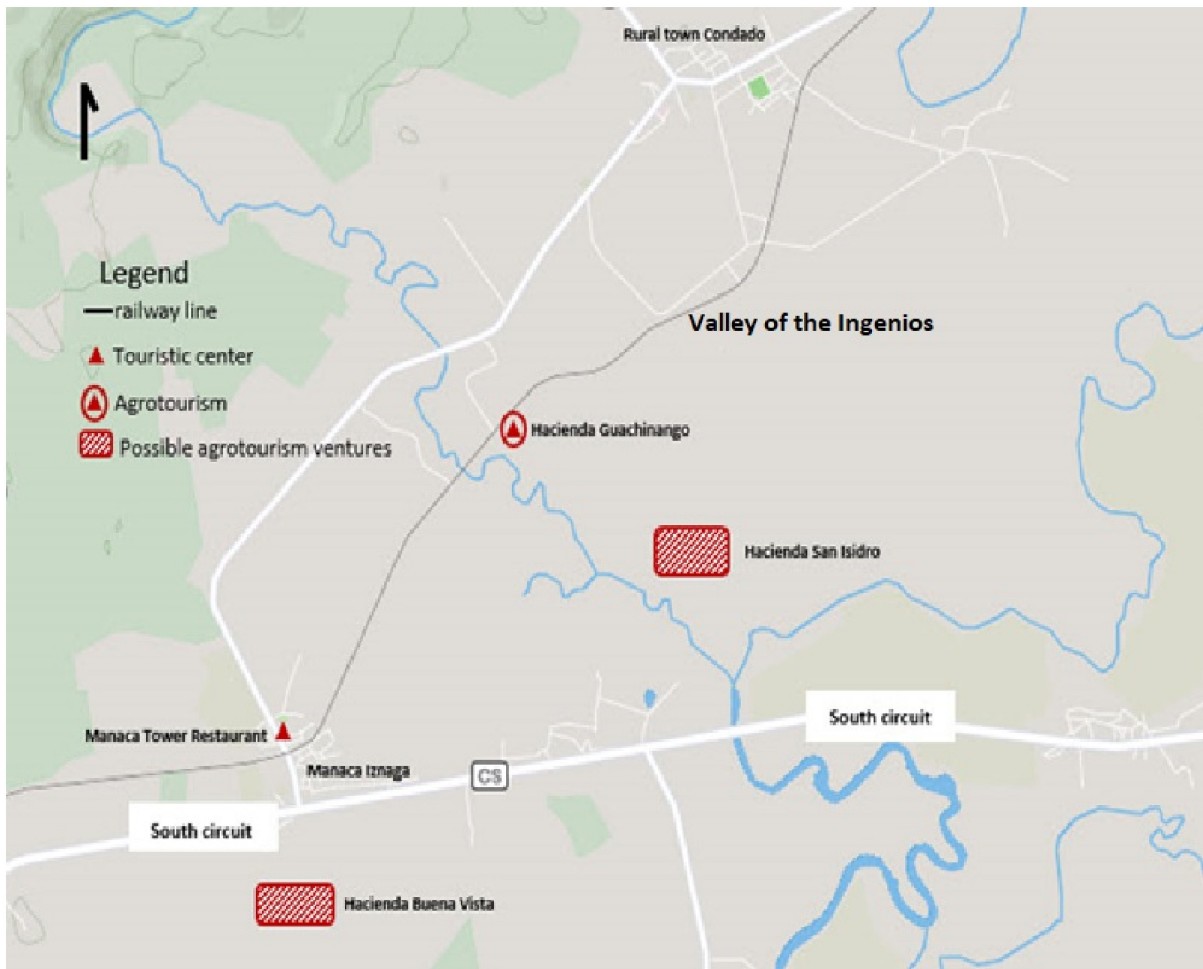

**Figure 1.** Location of Hacienda Guachinango. Source: own elaboration from [87].

The Hacienda is, in a social context, characterized by high levels of alcoholism, family conflicts, and crime. The main source of employment is agriculture, which does not cover the demand for work. The low wages lead to high precariousness of life. Poor families with incomes below the country's average, which is USD 1.90 per day, predominate [88]. The fundamental problems in the social sphere are in the poor condition of the roads, lack of an aqueduct network, inefficient public transport service, poor condition of sports areas, and poor public lighting. All this demonstrates the existing difficulties in the public services offered by the state. Considering the above, it can be said that Hacienda Guachinango is located in a disadvantaged area, in accordance with the definition of the most disadvantaged areas made by the Organisation for Economic Co-operation and Development (OECD), which is related to the results of the study carried out by [43] in Cuba. This is why it can be assured that the reactivation of tourism in the Hacienda and the opening of agrotourism in two other farms located nearby can contribute to the development and transformation of the socioeconomic situation of the area.

### 2.2. Methodological Framework

The research was carried out at Hacienda Guachinango from 2019 to 2021. The fieldwork made it possible to observe the environment within a radius of 4 km and to extend the study to nearby communities and two farms that present a similar situation and where agrotourism initiatives can be undertaken, as wages are low and farming families lead a precarious life that barely allows them to maintain low levels of agricultural production.

In order to contribute to the achievement of the proposed research objective, an exploratory and descriptive study based on the mixed paradigm (linking qualitative and quantitative) was projected on the basis of a qualitative analysis and description of the current situation of Hacienda Guachinango, with the aim of identifying the potential to satisfy the demand for agrotourism [89].

The synthetic analytical method was applied, and diagrams were drawn up and recorded in the research logbook. This required observation during fieldwork, as well as the application of interviews, surveys, the formation of focus group discussions, and brainstorming with actors directly related to the tourism resource. Descriptive statistics were used to process the results obtained. Figure 2 shows the diagram of the research methodology applied.

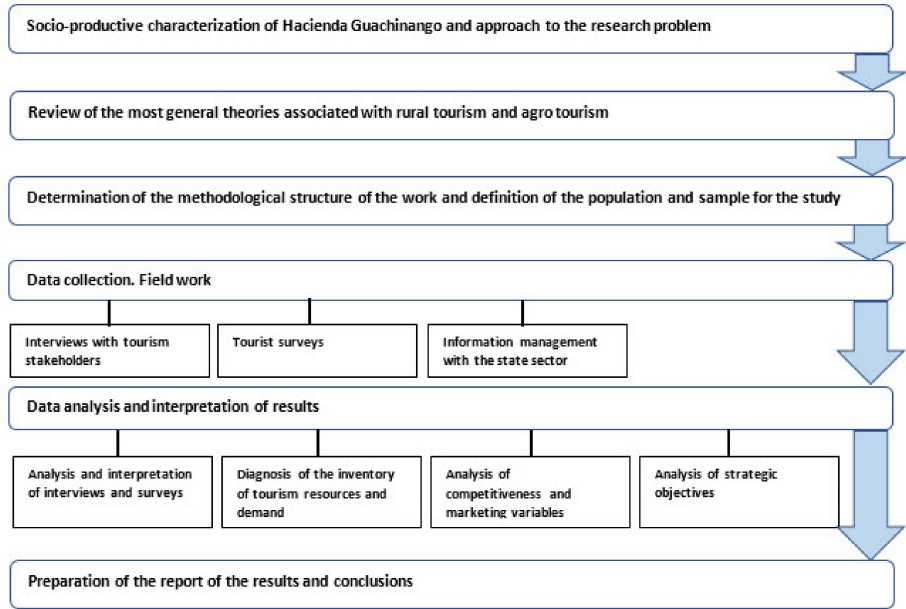

**Figure 2.** Diagram of the research methodology. Source: own elaboration based on [89].

To meet the stated objectives of the work, several data collection methods were designed and applied, including three semi-structured interviews, two surveys, and a focus group that included two sessions and a brainstorming session, all of which were necessary to collect information that was used in the elaboration of the resource inventories, the diagnosis, and to draw up the improvement plan for the enhancement of the agrotourism product at Hacienda Guachinango.

A non-probabilistic sample of 150 people was selected for the interviews, including a group of 15 workers from Hacienda Guachinango and 28 actors from the tourism sector at the municipal and provincial levels, a group of 23 local residents, and a third group of 84 clients. The interviews were aimed at ascertaining the strengths and weaknesses existing in the internal and external environment for the development of agrotourism and its potential. The instrument used was a semi-structured interview guide. A Microsoft Excel spreadsheet was used for the statistical management of the information and its qualitative interpretation.

Two structured opinion surveys were applied to a non-probabilistic sample of 95 people. In the first, all the workers (15 employees) of the Hacienda were considered; in the second, 80 national and international clients were considered to measure the level of satisfaction achieved with the tourist experience. The surveys were focused on measuring the agrotourism product and its quality, as well as the future potential to maintain an adequate position in the market and to enhance the value of the product. The instrument used was the survey questionnaire. For the workers, 15 structured questions were designed, and

for the tourists, 19 structured questions were prepared in Spanish, English, and German. A Microsoft Excel spreadsheet was used for statistical data management and calculations.

Both the interviews and the surveys were not administered at a single point in time, especially those taken with the tourists. It was necessary to visit the Hacienda several times between 2019 and 2020. In this sense, it was possible to provide information for the collection and analysis of data aimed at strengthening the qualitative criteria of the diagnosis on the performance of agritourism at Hacienda Guachinango.

The selection of non-probabilistic sampling responded to the fulfillment of the objective proposed for the work, since the aim was to obtain the cases that were of interest for the research and that offered a great wealth for the collection and analysis of the necessary information on the agrotourism product at Hacienda Guachinango [89].

In order to strengthen the qualitative component of the work, a focus group discussion was held in two sessions, one in 2019 and the other in 2020. The first included farm workers, while the second session considered tourism stakeholders from the provincial and municipal levels to learn and expand on ideas linked to the attention and satisfaction of tourists and their link with agricultural production, its results, and the participation of tourists in these activities.

The brainstorming was carried out with the purpose of enriching the information on the tourist product and the possibilities of diversifying the offer, based on the tourist resources of the Hacienda and the surrounding environment. A questionnaire was carefully prepared for this purpose. The questions were formulated in an open-ended way by means of a free-form discussion with all participants.

By means of observation, fieldwork, interviews, surveys, focus groups, and brainstorming, the results were triangulated, which allowed the structuring of a resource inventory as a dynamic tool [90,91] in order to establish a diagnosis related to the potential of Hacienda Guachinango to develop agrotourism, satisfy the preferences and expectations of tourists, and make decisions in this regard.

A marketing diagnosis was carried out in order to examine the four variables, namely, tourism product, price, distribution, and communication [92,93], as the main aspects for marketing planning. Based on the results, a mapping of the four key variables was developed in order to identify possibilities for improvement through properly planned strategic actions. A comprehensive diagnosis of the current situation was carried out, identifying the main problems and latent potentialities, strengths and weaknesses, as well as threats and opportunities. All this was done with the participation of the workers of the Hacienda and the actors of the tourism sector at the municipal and provincial level, which made it possible to organize the ideas in order of priority.

### 2.3. Data and Data Sources

Based on the diagnosis, a matrix of weaknesses and strengths (internal environment variables) and threats and opportunities (external environment variables) was created, with each assigned to the registered items in each category and the value of its impact on the tourist resource.

The classification of the strength items was as follows: high impact 4 points, moderate impact 3 points; for weaknesses: moderate impact 2 points, low impact 1 point; in the threat items: very high impact 4 points, high impact 3 points; and for the opportunities: regular impact 2 points, little impact 1 point.

Subsequently, each of the items corresponding to the internal environment was weighted with a relative value, the sum of which was equal to 1 ($\Sigma = 1$). The same procedure was followed with the items referring to the external environment.

The sum of the values obtained by multiplying the value of the impact assigned to each item of the internal environment by its weighting revealed the value of the competitiveness of the resource on a scale of 0 to 4 points. The location of both scores in the corresponding quadrant shed light on the strategy to follow.

The factors that have favored the development of agrotourism are not only on the supply side but also on the demand side [76]. Therefore, an analysis of the market was carried out according to supply and demand, as well as studies on image and satisfaction, to redesign the Hacienda Guachinango tourist product. There was also a study of the situation presented by the marketing variables, including product, price, distribution, and communication.

To analyze the behavior of the demand, the logical historical method was applied, accessing the statistical data registered in the Territorial Office of Tourism. The data quantified for this purpose were considered according to the system of indicators established for their records, such as inclusion of the technique of inventorying tourist resources and attractions, with their justification; a diagnosis of the product, price, distribution, and communication; market segmentation to determine the main segments of agrotourism and main markets that visit Hacienda Guachinango; the analysis of the surveys to determine to what extent the product constituted a special attraction during and after the COVID-19 pandemic; definition of the contribution of tourism to alleviate poverty and its contribution to depressed areas; and the application of a diagnostic technique that favors defining the strategy to follow, based on the variables of tourism in the territory.

For this purpose, two structured surveys were applied to a non-probabilistic sample of 95 people, including 80 international and national tourists and 15 employees of the Hacienda during the years 2019 and 2020. The first was focused on obtaining information on tourist preferences and the second to determine the degree of satisfaction of visitors who chose the Hacienda Guachinango tourism product.

For this, two surveys were applied to an intentionally selected sample of international and national tourists during the year 2021 and the first quarter of the year 2022. The first focused on obtaining quantitative information on the preferences of tourists and the second on determine the degree of satisfaction of visitors who chose the Hacienda Guachinango tourism product.

The classification of the items was carried out to determine the motivational factors and satisfaction of the tourists, with the application of an ordinal scale where the satisfied and unsatisfied demands on the site were determined. The following scale was employed: very satisfied 5, satisfied 4, neutral 3, dissatisfied 2, totally dissatisfied 1, with a margin of error = 4%, level of confidence = 95%, and origin of the issuing market (foreign and national) = 50%.

As an inclusion criterion, all tourists present at the time of the survey were considered, provided they gave their consent and had enjoyed the products offered at Hacienda Guachinango. As an exclusion criterion, foreign and national tourists who did not give their consent to participate were not included in the study as well as those newcomers who had not enjoyed the product offer and who, for obvious reasons, were unaware of it. For the application of the unstructured interviews, the interview guide was used, differentiating the topics for the tourists from those discussed with the workers of the Hacienda and the actors of the sector at the municipal and provincial level.

## 3. Results

### 3.1. Characterization of Tourist Resources

From the results of the semi-structured interviews with the selected sample, it was found that international and national tourists showed satisfaction with the agrotourism product on offer and referred mainly to the architectural and scenic value of the site, the natural and environmental benefits offered by the surroundings, as well as the possibilities of really getting to know the way of life in the Cuban countryside and understanding the real needs of the resident population.

The survey of the workers revealed the material limitations related to the logistics of providing a good service to tourists. They recognized the need to improve the quality of services and complementary offers, improve the infrastructure, and fix the access roads and signposting to and from the Hacienda.

Similarly, the need to develop strategic alliances with nearby tourist centeres, as well as the hostels in the city of Trinidad, and to take advantage of the opportunities offered by the tourist train, was raised. The need to adapt the offer to different market segments and to reconsider domestic tourism as a very important market was also raised.

The criteria of international and national tourists are positive and favorable. Aspects that need to be improved include the need for greater promotion and marketing of the site, better transport options for visiting neighboring communities, the improvement of access roads and signposting, and greater diversification of the offer.

The results of the focus group showed that there are criteria that coincide with the results of the surveys and interviews. The role of local governments and the need for strategic alliances between the public and private sectors emerged as a new element, as well as the need to encourage greater participation of communities and farmers in agrotourism projects, which could improve the living conditions of families and community development in the area.

The inventory as a register made it possible to organize the set of attractions that form part of the agrotourism product and to evaluate and rank the attractions. As a result, the different existing natural and cultural resources were evaluated and ranked, and their potential for attracting tourists was determined. The inventory of the delimited set of tourist resources in the area studied constituted the starting point for determining the reality on which to project the option linked to the modality of agrotourism.

The inventory of the delimited set of tourist resources in the studied area constituted the starting point to present the agrotourism modality. Figure 3 shows the result of the inventory of natural resources available for agrotourism in Hacienda Guachinango.

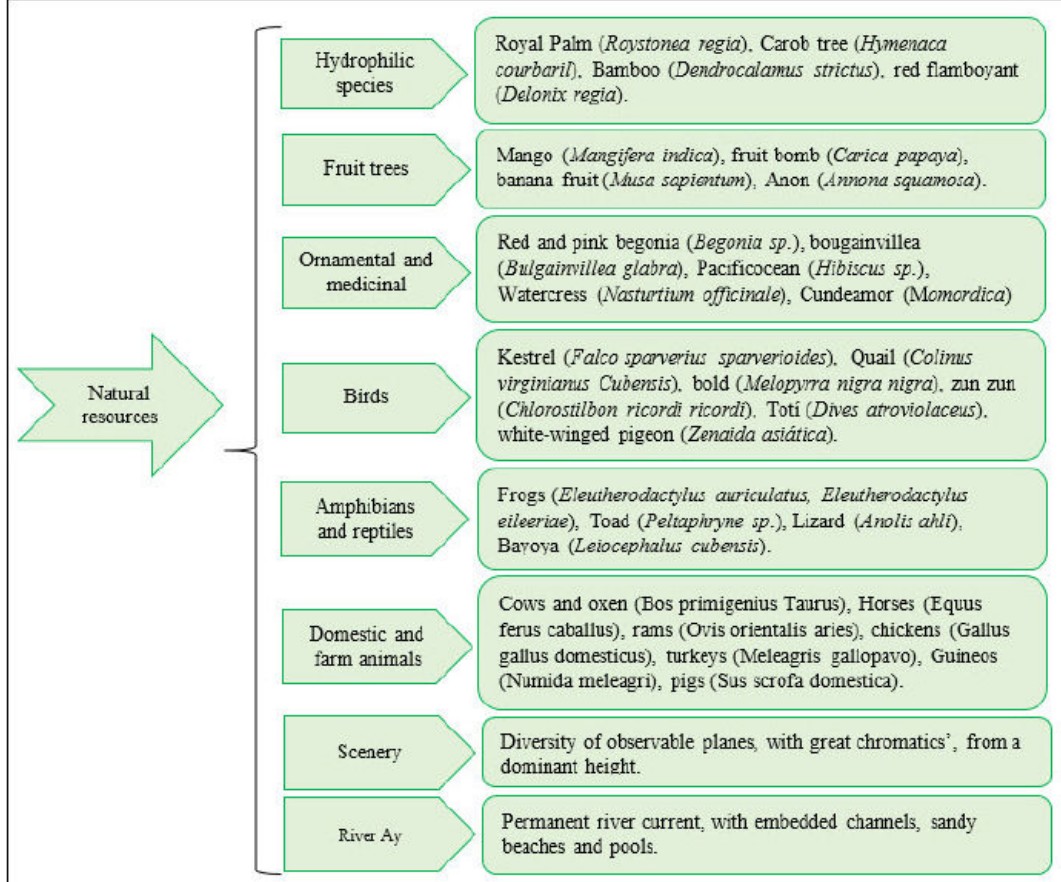

**Figure 3.** Inventory of natural resources available for agrotourism. Source: own elaboration based on the results of the inventory of resources in Hacienda Guachinango.

The inventory of the available natural resources reveals a significant potential for enhancing and improving Hacienda Guachinango's tourism products.

Figure 4 shows the result of the inventory of historical–cultural resources available for agrotourism in Hacienda Snapper.

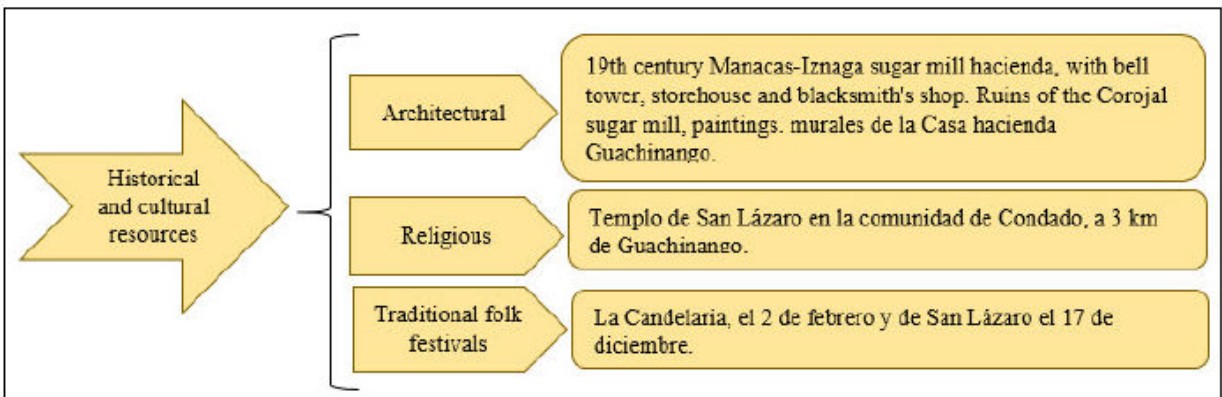

**Figure 4.** Historical–cultural resources available for agrotourism at Hacienda Guachinango. Source: own elaboration based on the results of the inventory of resources at Hacienda Guachinango.

The historical and cultural resources constitute another reserve of tourist attraction that is present in Hacienda Guachinango.

Figure 5 shows the result of the inventory of tourism infrastructure available for agrotourism at Hacienda Guachinango.

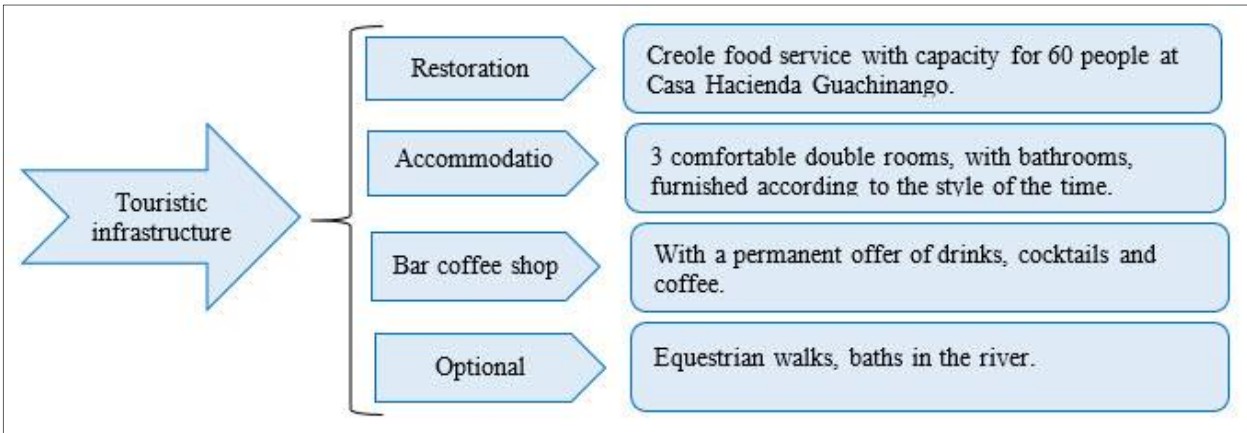

**Figure 5.** Tourism infrastructure available for agrotourism at Hacienda Guachinango. Source: own elaboration based on the results of the inventory of resources at Hacienda Guachinango.

The tourism infrastructure plays an important role in enhancing the value of tourism products. Hacienda Guachinango must work to improve its infrastructure, which can help to enhance the value of the agrotourism product on offer.

Figure 6 shows the result of the inventory of basic service infrastructure available for agrotourism at Hacienda Guachinango.

The application of the methods and techniques foreseen in the design of the Cuban tourism model was used to diagnose the product studied. Despite some difficulties detected in the resource inventory at Hacienda Guachinango, it can be seen that it has the necessary conditions for the development of agrotourism, which is corroborated by the analysis of the increase in demand in recent years. Attention needs to be paid to the difficulties and weaknesses detected.

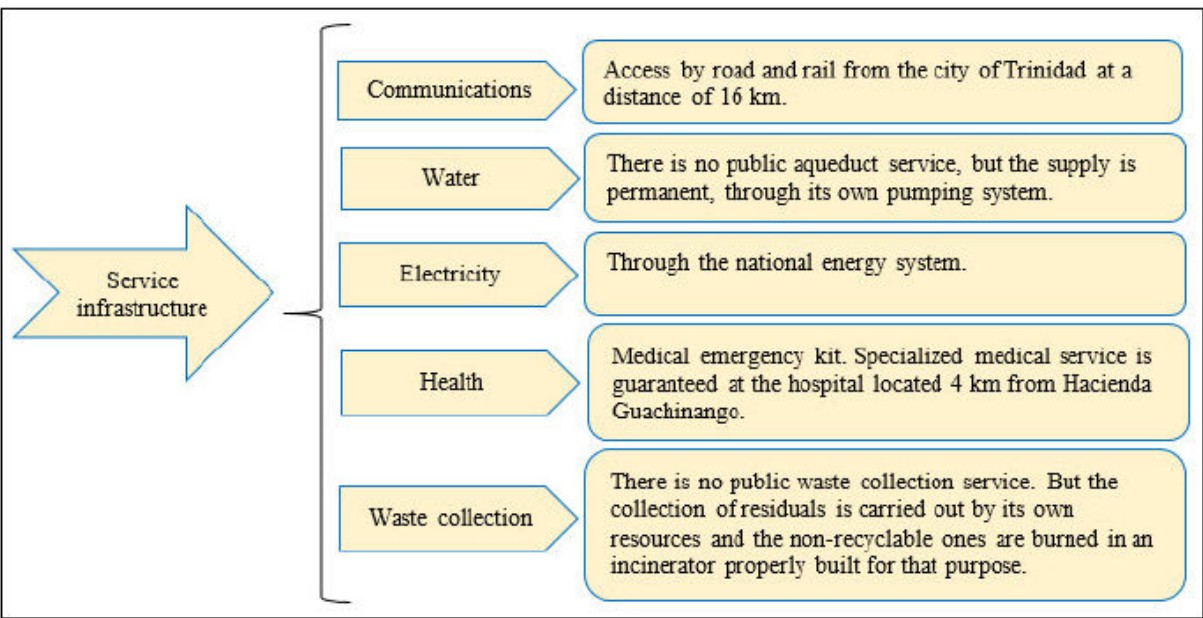

**Figure 6.** Infrastructure of basic services available for agrotourism in Hacienda Guachinango. Source: own elaboration based on the results of the inventory of resources in Hacienda Guachinango.

### 3.2. Marketing Study

In the marketing study, an assessment was made of the situation of the product, price, distribution, and communication variables. Figure 7 shows the result of the marketing study carried out at Hacienda Guachinango.

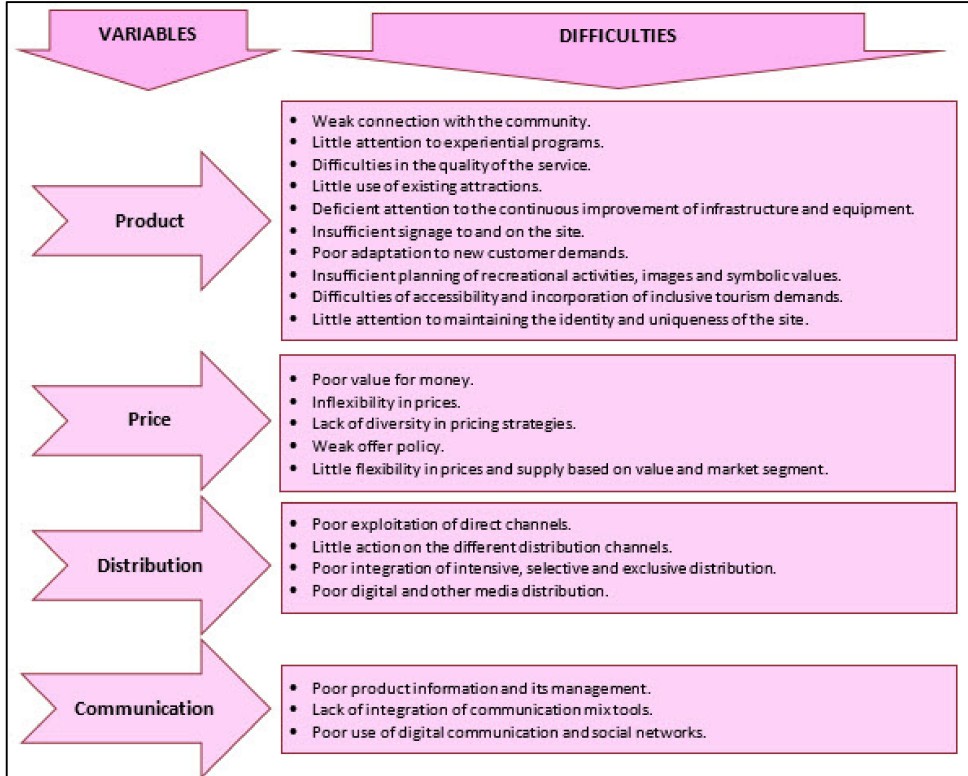

**Figure 7.** Results of the marketing study at Hacienda Guachinango Source: own elaboration based on the results of the marketing study at Hacienda Guachinango.

The marketing study at Hacienda Guachinango revealed a group of weaknesses that requires the formulation of a targeted strategy to raise the uniqueness and authenticity of the tourist product in the face of demand.

### 3.3. Analysis of Tourist Demand

Table 1 shows the behavior of tourist demand and economic income between the years 2012 and 2020 at Hacienda Guachinango.

**Table 1.** Behavior of demand and economic income.

| Indicators | 2012 | 2017 | 2018 | 2019 | 2020 |
|---|---|---|---|---|---|
| Tourists from travel agencies | 204 | 1038 | | 658 | 972 |
| Income | 1603.00 | 6302.00 | | 5270,00 | 7777.00 |
| Income Pax moved Agencies | 7860.00 | 6070.00 | Closed for remodelling | 16,100.00 | 28,800.00 |
| Individual tourists | 4633.00 | 1910.00 | | 12,505.00 | 18,481.00 |
| Income | 21,615.00 | 5950.00 | | 100,040.00 | 147,848.00 |

Source: own elaboration based on data from [67].

The tourism management of the product experienced sustained economic growth in demand in the last decade despite the impact of the COVID-19 pandemic in March 2020, which significantly affected the performance of other tourism products in the country. During the sanitary measures, tourists looked for places of recreation and relaxation in natural places where they could enjoy fresh air, away from urban settings where the disease wrought the most severe damage.

### 3.4. Analysis of Competitiveness and Attractiveness of the Tourist Product

The marketing diagnosis provided a detailed understanding of how Hacienda Guachinango is visited by its public. This is an essential aspect for marketing planning. As a result, a mapping of the four marketing variables was obtained in order to identify possibilities for improvement through future strategic actions. A comprehensive diagnosis of the current situation was made, and the main problems and potential opportunities were identified, as well as strengths, weaknesses, threats, and opportunities through brainstorming. List reduction and weighted voting allowed the ideas to be organized in order of priority. The main issues affecting the marketing variables were identified and are summarized in Table 2 and Figure 8.

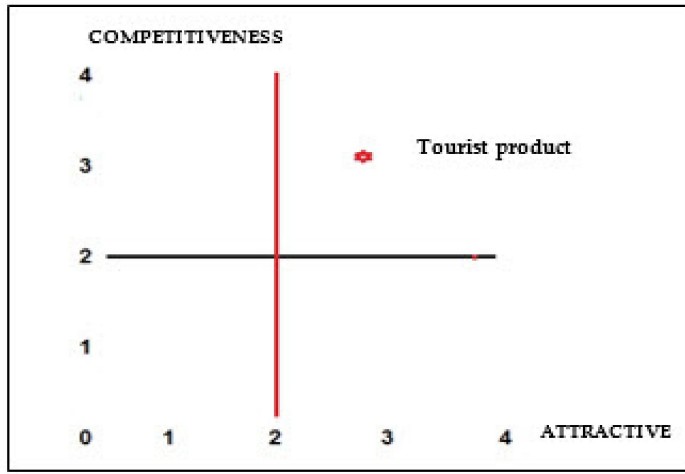

**Figure 8.** Competitiveness and attractiveness of the tourism product. Source: own elaboration based on the results of the SWOT analysis.

**Table 2.** Strengths and weaknesses of the internal environment and threats and opportunities of the external environment of the product.

| Around | SWOT | Items | Impact (I) | Weighing (W) | I × W |
|---|---|---|---|---|---|
| Internal | Strengths | Location in a heritage area | 4 | 0.2 | 0.8 |
| | | Building of great architectural value | 3 | 0.1 | 0.3 |
| | | Proper control environment | 3 | 0.1 | 0.3 |
| | | Prepared workers and experts in providing quality tourist services | 3 | 0.1 | 0.3 |
| | | Sufficient areas to promote agrotourism activities | 4 | 0.2 | 0.8 |
| | weaknesses | Insufficient product promotion | 2 | 0.08 | 0.16 |
| | | Incomplete livestock module | 2 | 0.08 | 0.16 |
| | | Insufficient equipment and supplies | 1 | 0.06 | 0.06 |
| | | Few options related to agrotourism | 2 | 0.08 | 0.16 |
| | | Σ | | 1 | 3.04 |
| External | threats | Appearance of other competing products in the Valley | 4 | 0.2 | 0.8 |
| | | Access roads in poor condition | 3 | 0.15 | 0.45 |
| | | Affectations of the native flora and fauna | 3 | 0.15 | 0.45 |
| | | Few links between agricultural entities and the tourism sector | 4 | 0.2 | 0.8 |
| | Opportunities | Growing international interest in visiting the tourist pole | 2 | 0.1 | 0.2 |
| | | High level of occupancy of hotels and hostels in the city | 1 | 0.05 | 0.05 |
| | | State decision to support development projects in the Valley | 2 | 0.1 | 0.2 |
| | | Σ | | 1 | 2.9 |

Source: Own elaboration based on the results of the field work.

The results of the strategic diagnosis of the product in the strengths, weaknesses, threats, and opportunities (SWOT) matrix carried out at Hacienda Guachinango are shown in Table 2.

The values obtained for the competitiveness of the product (3.04), as well as for the degree of attraction it exerts (2.90), show that a progressive strategy must be continued betting on product growth, as shown in Figure 8, for which the option of incorporating the values of agrotourism, taking advantage of the resources available in its environment, is decisive.

In order to determine the preference of tourists for the products offered at the Guachinango farm, a nominal scale was applied. The results are reflected in Table 3.

Figure 9 shows the general behavior of the preferences of tourists coming from abroad in relation to domestic tourism.

The work revealed that foreign tourists are usually more demanding in terms of the preferences of the tourist products offered by Hacienda Guachinango. There is a correlation between the hygiene and comfort of the location. Foreign tourists shows more interest in aspects related to security; the level of information on products and services; health coverage; application of biosecurity protocols; coverage of banking care; knowledge of the history, culture, and traditions of the countryside; exploration of the way of life of the Cuban peasant; maintaining exchange with local population, which carries out sustainable practices; application of agroecological techniques; participation in agricultural activities; application of innovation; and physical exercise.

**Table 3.** Preference of tourists for the products offered.

| Products | Foreign Tourist Satisfaction (Average) | Satisfaction of Domestic Tourists (Average) |
|---|---|---|
| High security | 5.0 | 4.2 |
| Level of information about the products and services on offer | 4.7 | 4.3 |
| Health coverage (speed of medical care and quality) | 4.8 | 3.4 |
| Hygiene of the place | 5.0 | 5.0 |
| Comfort | 5.0 | 5.0 |
| Implementation of biosecurity protocols | 4.8 | 4.1 |
| Coverage of banking and currency exchange | 4.7 | 3.0 |
| Fair price (value for money) | 5.0 | 5.0 |
| Knowing the benefits offered by agritourism practices | 4.5 | 4.7 |
| Contribution of the activity to revitalise underprivileged areas | 4.6 | 4.8 |
| Knowledge of the distribution of the benefits generated by tourism | 4.4 | 4.9 |
| Contribution to local development | 4.3 | 4.7 |
| Support that tourism receives from the state sector | 4.1 | 4.8 |
| Knowledge of the relationship between the public and private sector. | 3.9 | 4.7 |
| Knowledge of the history, culture and traditions of the countryside. | 4.7 | 4.5 |
| To explore the way of life of the Cuban peasants. | 4.8 | 3.8 |
| Enjoying peasant or Creole food. | 4.7 | 4.8 |
| Exchange with local people. | 4.8 | 4.4 |
| Sustainable practices applied | 4.8 | 4.3 |
| Application of agro-ecological techniques | 4.8 | 4.3 |
| Participate in agricultural activities | 4.8 | 3.6 |
| Application of innovation | 4.5 | 4.2 |
| Carrying out physical exercise. | 4.4 | 4.3 |
| I would recommend your visit | 4.5 | 4.4 |

Source: own elaboration based on the results of the survey of the preferences of tourists.

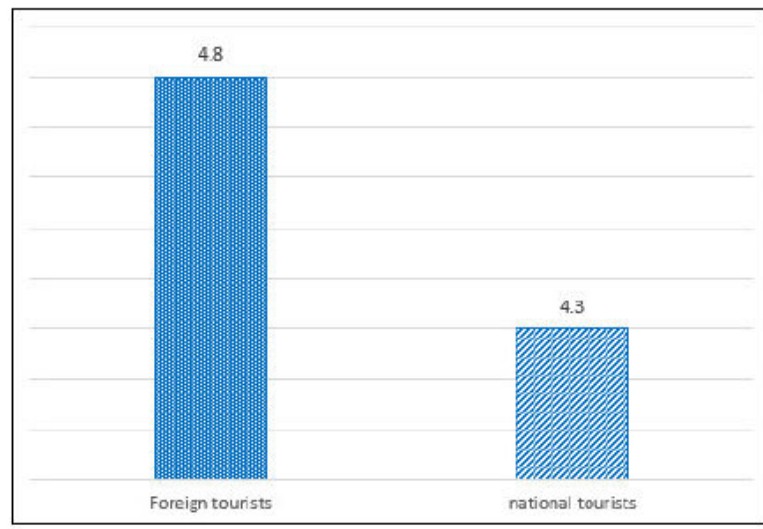

**Figure 9.** General situation of the preferences of foreign and domestic tourists for the tourist product. Source: own elaboration based on the results of the survey of the preferences of tourists.

## 4. Discussion

The inventory of the delimited set of tourist resources in Hacienda Guachinango was the starting point to determine the agrotourism option, which led to a dynamic management system for tourism development, as discussed in [90,91]. It corroborates the existence of diversity of resources with the required potentialities for the enhancement of the agrotourism model as a product that is growing and that has a high demand and excellent prospects in the market, as referred to in the work carried out by [63,64]. It is argued that agrotourism can contribute to minimizing the effects of the crisis with respect to other sectors, given the character and characteristics of the product, and provides an opportunity to reposition itself in the market, which is corroborated in the research [38]. The authors of [65,66] point out that rural tourism is a productive form of tourism that, as a tourist activity, shows the nature, culture, gastronomy, routes, and accommodation of rural areas.

The building infrastructure at Hacienda Guachinango is old and deteriorated. The road and access roads are in poor condition. The transport service is precarious and makes access to the site difficult. All these are barriers that affect the competitiveness of agrotourism. The authors of [47–49] point out that agricultural activity has experienced a significant reduction in investment and that this is related to the deficient infrastructure for basic services such as electricity, healthy water supply and crop irrigation, low access to technological communication services, and inadequate and in some cases no public aid.

In spite of this, it can be seen that the farm has the necessary conditions for the development of agrotourism. It is necessary to pay attention to the difficulties and weaknesses detected, as suggested by the authors of [48–50].

The work revealed that the main market segments by geographical areas are Canada, Germany, Italy, and the national market. These arrive individually or through the excursion offered by tourist groups. Tour or circuit tourism predominates, with an average stay of one day.

The product has experienced a sustained growth in demand as a result of tourism management over the last decade, despite the impact of the COVID-19 pandemic in 2020. The measures of isolation and restriction of movement negatively impacted other tourist products in the country. In the health situation created by the disease, tourists preferred places where they could enjoy fresh air, away from urban centers where the pandemic manifested itself with particular harshness. This is confirmed by the World Tourism Organisation [47].

Some authors such as [31,32] state that adequate and permanent monitoring of tourist preferences plays a determining role in the valorization of the agrotourism product. Table 4 shows the results of the motivational satisfaction of the tourists who visited the Hacienda Guachinango.

When it comes to the satisfaction of tourists, foreigners demonstrate a higher level of satisfaction than nationals. There is agreement in the high security offered by the place. Foreign tourists tend to perceive more satisfaction in products related to the level of information on products and services; health coverage; hygiene; comfort; application of biosecurity protocols; knowledge of the economic and non-economic benefits offered by agrotourism practices; the contribution of the activity to boosting less favored areas; knowledge of the distribution of the benefits generated; contribution to local development, knowledge of the history, culture, and traditions of the countryside; exploration of the way of life of the Cuban peasant; enjoyment of peasant or Creole food; sustainable practices that are applied in the place; application of agroecological techniques; and participation in agricultural activities, in line with what has been suggested by the authors of [6,8,19,20].

For all these reasons, foreign tourists would recommend other people to visit the place and express their intention to revisit and enjoy the tourist products at Hacienda Guachinango. National tourists would recommend other people to visit the place, but not all of them say that they would return to the place.

**Table 4.** Level of perception of tourists regarding their preferences for the selection of the agrotourism product. Differences between foreign tourists and domestic tourists.

| Products | Foreign Tourist Satisfaction (Average) | Satisfaction of Domestic Tourists (Average) |
|---|---|---|
| High security | 5.0 | 5.0 |
| Level of information about the products and services on offer | 4.7 | 4.0 |
| Health coverage (speed of medical care and quality) | 4.8 | 4.5 |
| Hygiene of the place | 5.0 | 4.4 |
| Comfort | 5.0 | 4.6 |
| Implementation of biosecurity protocols | 5.0 | 4.7 |
| Coverage of banking and currency exchange | 4.2 | 4.0 |
| Fair price (value for money) | 5.0 | 4.3 |
| Knowing the benefits offered by agritourism practices | 4.5 | 4.2 |
| Contribution of the activity to revitalise underprivileged areas | 4.7 | 4.5 |
| Knowledge of the distribution of the benefits generated by tourism | 4.5 | 4.1 |
| Contribution to local development | 5.0 | 4.5 |
| Support that tourism receives from the state sector | 4.3 | 4.0 |
| Knowledge of the relationship between the public and private sector. | 4.0 | 4.3 |
| Knowledge of the history, culture and traditions of the countryside. | 5.0 | 4.0 |
| To explore the way of life of the Cuban peasants. | 5.0 | 4.2 |
| Enjoying peasant or Creole food. | 4.6 | 5.0 |
| Exchange with local people. | 5.0 | 3.6 |
| Sustainable practices applied | 4.5 | 4.4 |
| Application of agro-ecological techniques | 4.8 | 4.3 |
| Participate in agricultural activities | 5.0 | 4.0 |
| Application of innovation | 4.3 | 4.1 |
| Carrying out physical exercise. | 5.0 | 4.6 |
| I would recommend your visit | 5.0 | 5.0 |
| I would visit the place again | 5.0 | 4.3 |
| Average | 4.7 | 4.3 |

Source: own elaboration based on the results of the survey of the preferences of tourists.

All this demonstrates a presumed differentiation in terms of treatment and care between foreign and domestic tourism. It is recommended that the tourist actors of Hacienda Guachinango understand the importance of maintaining the same level of attention and satisfaction for domestic tourists.

The research revealed that, there is a diversity of definitions for the criteria related to tourism competitiveness, and there is no consensus among the authors who have studied the subject. Among them, we can mention [93,94], although it may be possible to find correlations in the interpretation of the different aspects that include the definition of [95], which stands out regarding a competitive destination referring to tourist spending through the offer of memorable and satisfactory experiences that are profitable for the destination by improving the well-being of the resident population and preserving their natural capital for future generations, aspects that are consistent with the paradigm of sustainable tourism development [96].

It is argued that an adequate strategy focused on the study of demand and the valuation of the tourist product can promote the modality of agrotourism as a viable option for

the improvement of the socioeconomic conditions of the family dedicated to agricultural production, which can reduce the differential marginality between urban and rural life, reduce migration from the countryside to the cities, increase adequate jobs, and minimize rural precariousness in a scenario of resignification of natural potentialities, focused on the broader concept of environmental sustainability [97,98].

Some authors such as [28–30] demonstrate in their work the importance of considering the behavior of tourism demand. They propose some strategies to take advantage of the agrotourism potential. They recommend analyzing the offer and its possible components, the design of local projects for the different segments, the adjustment of the offer to the demand, the promotion and publicity of the tourist product, the need to strengthen training through courses for farmers, the implementation of technical and business training programs, the implementation of studies in external facilities to appreciate the experiences in other similar projects, the promotion of traditional polycultures, the reconversion of agrotourism with a view to the future, and the improvement of accesses and communications.

The strategy involves deep knowledge of the market and its competitors to determine the ability to cope by identifying the possible risks. This allows an adequate selection of key customers to venture into new segments taking advantage of the opportunity of the health crisis marked by the effects of COVID-19.

The product must continue to make its strengths visible, which means being in a high-value heritage area and having sufficient areas to promote the development of agrotourism and opportunities to capitalize on the growing international interest in the product.

Efforts should be redoubled to minimize the present weaknesses and reverse the situation related to the limited options, in line with what was stated by [32]. For this, it is necessary to diversify the supply based on the unmet needs in demand, improve promotion and equipment, complete the livestock module, mitigate the threats of the emergence of new competing products and the lack of linkage of agricultural entities with the tourism sector, and improve access and strengthen strategies for the preservation of the environment, as suggested by the authors of [70,71].

The work revealed that there are possibilities to expand the offer so that it increases both the competitiveness and the attraction of the product through the incorporation of modalities related to the participation of tourists in the clearing of land by animal traction (oxen); realization of cultural attention to the various crops according to the season, traditional customs, and habits; feeding of domestic species and poultry; milking of cattle and goats using traditional methods; excursions to nearby sites of sociocultural interest; participation of tourists in environmental management and education programs for sustainable development; knowledge of good practices of agrotourism; and the possibilities of cultural exchange with peasant families and the local population. All this integrated in the development of a comprehensive program of interpretation of the natural and cultural heritage can achieve the communication of its significance and relevance, in line with what was pointed out by [69].

This imposes the need to apply marketing in a creative and innovative way, to be proactive with the ability to get ahead of market competitors. The authors of [92,93] provide elements in their works that show that marketing constitutes a main element to enhance the value of the tourism product by generating business and opportunities, which serve to raise awareness, motivate, convince, and put each of the values of a given destination in the mouths of potential users, visitors, or travelers.

The fieldwork carried out revealed the existence of a group of difficulties in the marketing variables, as reflected in Figure 7. This led to the design of a group of strategic objectives that must be met to reduce the weaknesses and problems detected. Figure 10 shows the strategic objectives for each marketing variable.

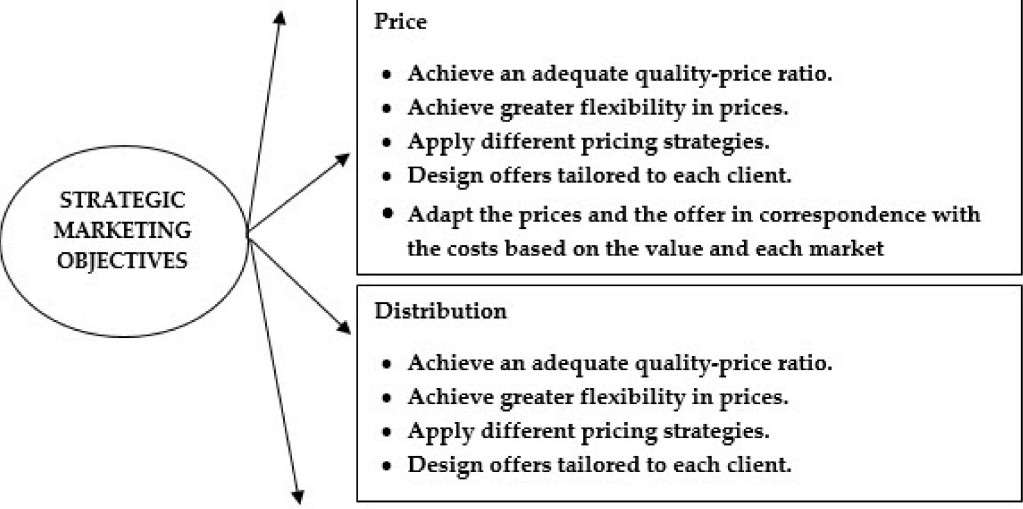

**Product**

- Promote greater connection with the community.
- Develop experiential programs.
- Improve the quality of service.
- Increase the promotion of existing attractions.
- Continuously improve infrastructure and equipment.
- Improve signage to and on the site.
- Adapt services to new customer demands.
- Incorporate more recreational activities, images and symbolic values.
- Improve accessibility and incorporate the demands of inclusive

**Price**

- Achieve an adequate quality-price ratio.
- Achieve greater flexibility in prices.
- Apply different pricing strategies.
- Design offers tailored to each client.
- Adapt the prices and the offer in correspondence with the costs based on the value and each market

**Distribution**

- Achieve an adequate quality-price ratio.
- Achieve greater flexibility in prices.
- Apply different pricing strategies.
- Design offers tailored to each client.

STRATEGIC MARKETING OBJECTIVES

**Communication**

- Improve product information and its management through tourist information offices (INFOTUR).
- Integrate the tools of the communication mix to achieve marketing objectives.
- Promote digital communication by promoting work with social networks.

**Figure 10.** Strategic objectives for each marketing variable Source: own elaboration from the marketing study.

## 5. Conclusions

The fulfillment of the objectives set for the work made it possible to characterize the agrotourist product of the Guachinango Hacienda in the Valle de los Ingenios in the municipality of Trinidad, which made it possible to understand its significance, relevance, and perspectives for the development of the agrotourist modality in the context of rural tourism, by integrating into it the historical, cultural, and patrimonial values treasured for more than four centuries of existence. To this must be added the values of the daily work of the small family dedicated to agricultural production, with the possibility of the tourist being able to share unprecedented experiences in an environment of family production, cultural exchange, and the positive synergies of agricultural production.

The development of the objective of the work revealed that the agrotourist enterprise of Hacienda Guachinango has a staff of 15 workers, of which 2 are directly responsible for attending to tourism. The rest are dedicated to agricultural production and the care of large and small livestock. The income derived from the tourist activity is added to the general account of the farm, which improves the income of the staff, achieves greater equity, and



improves the living conditions of the people. Another advantage is the reinforcement of the amount of investments of the farm in order to achieve higher and sustainable production. In this way, the theoretical study of the agrotourism modality at Hacienda Guachinango allowed us to know that the tourism enterprise does not affect the social destination of agricultural production but, on the contrary, strengthens it in correspondence with the criteria provided by [5,7–12,14], in contrast to what was proposed by the authors of [15,16]. From this stems the relevance of the theoretical study of the agrotourism modality in Cuba as a new phenomenon that stems from the emerging need to develop tourism as an activity that guarantees the economic progress of the country.

The analysis of the socioeconomic context of Hacienda Guachinango revealed that there is a high level of alcoholism, social conflict, and petty crime and few employment opportunities, with low wages. With the exception of electricity, the area does not have the rest of the basic services, and all this describes a climate of precariousness and poverty in the peasant families, so it can be seen that the characteristics that allow the area where the Hacienda is located to be classified as less favored are present. This corresponds to the World Tourism Organisation [47] and is corroborated by the results of the study carried out by [43] in Cuba, despite the fact that the Valle de los Ingenios was declared a World Heritage Site by UNESCO in 1988.

It is concluded that in Hacienda Guachinango there are the conditions and the natural potential required to revitalize agrotourism as an activity that is born in the heart of the peasant family, capable of generating economic income to reduce the precariousness of life in the countryside and redefine the natural, historical, social, and productive values with the aim of increasing agricultural production and reducing migration to the cities. The methodology applied and the experiences derived from the research can be applied in similar rural contexts.

The work revealed that there is a wide range of resources and attractions that could be used to establish a post-COVID-19 agrotourism reactivation model for rural areas within the framework of sustainability. It was discovered that in the area surrounding Hacienda Guachinango, there are two other farms with real possibilities for agrotourism business. The results of the research can serve as a theoretical starting point for further study of the modality and be applied in the two farms mentioned above, as well as in other similar environments in the country and Latin America, where there are many peasant families who daily face the precariousness and high cost of living that they have to pay for the fact of dedicating themselves to small-scale agricultural production in a rural area. All of this in spite of having natural, historical, and cultural riches similar to those found in Hacienda Guachinango.

The results of the surveys carried out with foreign and national tourists showed that Hacienda Guachinango has a high motivational attraction for the enjoyment of the tourist products on offer. Resources that should be adequately exploited in accordance with what is stated by [34].

The research revealed the competitiveness of Hacienda Guachinango and the scope to develop the tourism modality of agrotourism, given its current capacity and potential to develop competitive advantages over its competitors by obtaining a prominent position in terms of the quality and variety of tourism products that are unique in their environment.

The systemic analysis of the different applied research methods identified the real perspectives of agrotourism in Hacienda Guachinango as a dynamic element of agricultural production to improve the socioeconomic conditions of small peasant families that can benefit collaterally with the additional income generated by the tourist modality in less favored areas.

However, the theoretical study of agrotourism is not free of limitations. Among them is the lack of experience with this recently emerging tourism modality, which does not have sufficient theoretical studies that address the subject in a comprehensive manner.

The study of agrotourism is inadequately appreciated by the tourism sector as a marginal problem of a local nature, which does not require priority attention for research

and development, as well as the promotion of the tourism value of the resources that exist in the rural environment.

The lack of investment by the State to improve roads, transport services to and from rural communities, poor coverage of Internet communication, low quality and reliability of electricity services, scarce access to bank credit at low interest rates that would allow facilities to be equipped for tourism, and lack of commercial networks to guarantee the inputs required for visitor services are all limitations that affect the generalization of the agrotourism modality, in accordance with what has been stated by [33].

Particularly in Cuba, the poor working relationship between the public and private sectors is another limitation to the study of agrotourism. In the country, the opening of businesses to the private sector is a recent phenomenon that has not yet achieved the necessary experience to provide a wide opening to private enterprises, which is reflected in weaknesses in promoting public policies aimed at benefiting private business.

From the internal point of view, people directly involved in the management of the tourism activity lack preparation, a situation that constitutes a priority and that should be resolved through alliances with the tourism sector aimed at the training of these people.

There is a lack of understanding of the importance of national tourism compared to foreign visitors, as can be seen in the results of the survey on satisfaction with the enjoyment of tourism products, where foreign tourists have a much higher level of demands and requirements for the consumption of agrotourism products compared to national tourists. It was found that the actors in the tourism activity do not treat national tourists in the same way as foreign tourists.

**Author Contributions:** Conceptualization, N.P.E. and A.V.P.; methodology, A.V.P. and N.P.E.; formal analysis, N.P.E. and A.P.N.; investigation, N.P.E. and A.P.N.; data curation, N.P.E. and A.V.P.; writing—original draft preparation, N.P.E. and A.V.P.; writing—review and editing, A.V.P. and A.P.N.; visualization, N.P.E. and A.V.P.; supervision, N.P.E.; project administration, N.P.E. All authors have read and agreed to the published version of the manuscript.

**Funding:** This research received no external funding.

**Institutional Review Board Statement:** Not applicable.

**Informed Consent Statement:** Not applicable.

**Data Availability Statement:** Data can be provided upon request from the corresponding author.

**Acknowledgments:** The authors thank the authorities of the Ministry of Tourism and the owners of private tourist establishments in Hacienda Guachinango and Valle de los Ingenios, who gave their consent and collaborated in carrying out the work.

**Conflicts of Interest:** The authors declare no conflict of interest.

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
