# Peer review of "Rural Agrotourism Development Strategies in Less Favored Areas: The Case of Hacienda Guachinango de Trinidad"

_agriculture, doi:10.3390/agriculture12071047_

Round 1

Reviewer 1 Report

Introduction: Authors start with a statement that “Agrotourism can become an economic solution for the rural family…”, providing numerous references. Despite the wide variety of sources used, authors should give a clear link of why these studies are relevant. For example, such phrases as “The social, economic, and ecological benefits are also determined. Practical measures for farm management in the interest of increasing production are included” or “Problems and causes that prevent its development are identified, which can affect the development of tourism alongside agriculture and sustainability” are too general. Why do you mention that? What are scholars saying about developing agrotourism in less favored areas?

Also, what is a “less favored area”? Is a Cultural Heritage of Humanity site 105 by the United Nations Organization for Education, Science and Culture part of it?

Moreover, please consider the limitations of agrotourism in economic development of territories. There are findings that would contradict this statement as well.

As mentioned by the authors, many studies have been held in Cube (e.g. sources 23,24,25,26,27,28,29), therefore, it is important to showcase the background. What are the findings of previous studies?

Conclusions should not be part of Introduction (It is concluded that in the Hacienda Guachinango, the conditions required to promote agrotourism are met as an activity that is born within the rural family capable of generating economic income that makes it possible to reduce the precariousness of life in the countryside and redefine the natural, historical, social, and productive values with the aim of increasing agricultural production and reducing migration to the cities. The applied methodology and the experiences derived from the research can be applied in similar rural contexts.).

Theoretical delimitation: Put this section after the Introduction and put it in the context – why it is important to specify the agritourism type of rural tourism in this research?

Materials and Methods: Description of the study area and a map should provide better understanding of the tourism capacity (settlements, tourist sites, etc.). Also, what is the socio-economic status of the area? Why do you consider it as less favored with a need (and capacity) to develop other types of employment?

Overall, this section is too descriptive and blurry. Please indicate how many interviews, surveys, and discussion groups did you have; How and when did you take them exactly? Who was participating exactly? A reader needs to understand the sequence of methodological actions and the source of materials you have used. It remains unclear it is current state.

Results: Could you visualize tourist resources on a map? In is unclear how was the inventory of resources presented in Figures 3-6 been done. How was the marketing study done? For example, who suggests there is “Poor value for money”?

What data proves the following statement: “During the sanitary measures, tourists looked for places of recreation and relaxation in natural places where they could enjoy fresh air, away from urban settings where the disease wrought the most severe damage.”?

It remains unclear of what data and procedures were used to evaluate the role of agrotourism in developing less favored areas.

Author Response

Good afternoon.

We would like to thank you for your comments and recommendations, which have been taken on board in their entirety and have led to improvements in the work.

During the review and revision, it was necessary to add new content, which you will be able to check through the change control.

The content was incorporated into the article that demonstrates a clear link to why the studies are relevant, based on the dichotomy between criteria that approve of agrotourism as a form of development and other criteria that do not:

Some authors such as [1,3,4,5,6] state that the diversification of enterprise portfolios on farms dedicated to agro-tourism increases income and alleviates the economic hardship of families. They point out that especially in the United States of America, agritourism has increased in recent years.

Agro-tourism can generate environmental and socio-cultural benefits. An evaluation of the sustainability of farms quantified superior results compared to other agricultural facilities. Data evaluated on 873 farms in the United States of America with a diversified business portfolio showed that agritourism farms are more sustainable than their counterparts by producing environmental, socio-cultural, and economic benefits for farm labor, families, and society [1].

Authors Busby and Rendle [3] point out that the traditional approach to agrotourism has evolved in recent years from a complementary commercial activity to a tourism model in its own right, in line with authors such as (Denman, 1994a; Clarke, 1996a; Hjalager, 1996) who predict further growth in demand for this tourism product. It is emphasized that from a psychological point of view, it constitutes a different attraction for the tourist by sharing the experiences of life in a country house with accommodation, breakfast and enjoying family-style hospitality (Voase, 1995, p. 160).

There are scholars who raise other criteria related to the modality of agro-tourism and point out that further theoretical research is required to assess the extent to which this is a valid assumption. (Worth, 1997) noted in his studies that a farmer who started the tourism business in 1989 considers that it is now more profitable than his traditional farming activities. (Morris and Romeril, 1986) point out that agricultural activity cannot be subordinated to tourism, as it loses its productive essence and with it the productivity of food. 

The above-mentioned allows us to see the relevance of the theoretical study of the concept and practical criteria of the agrotourism model, which is corroborated in the ideas of (Oppermann, 1995) when he pointed out in his studies that agrotourism still lacks a comprehensive body of knowledge and an adequate theoretical framework, due to problems with the definition. (Deegan & Dineen, 1997) point out that the term is sometimes used interchangeably with rural tourism. (Page and Getz, 1997) point out that there are not enough studies related to the growth and development of agrotourism and the need to understand the dynamics of such business ventures.

For example, in sentences such as "Social, economic, ecological and social benefits are also determined, the following text was added to the work:

Social, economic, ecological, and social benefits are also determined, in that sense meeting the food needs for the family and dedicating the surplus for economical marketing. The income is used for health, education, clothing, housing, recreation, and to raise the quality of life and the socio-economic level of the farmer by strengthening the farm with a view to the future of the family.

Ecological management preserves the natural balance of the soil, maintains soil fertility, reduces erosion, and preserves biological populations. Crops are healthier and consumers eat healthier food. Social benefits include job stability, individual and family well-being, and self-esteem in the context of increased social and community participation. The reduction of poverty and marginalization, the preservation of ancestral and cultural values. The strengthening of human values. The promotion of constant self-improvement and creativity of the farmer and the flourishing of handicraft activities developed within the community framework [19].

Practical measures for farm management are included in the interest of increasing production, with emphasis on the introduction of innovations that are low cost for the farmer such as the implementation of organic manure, seed selection, and organic fertilization, in order to increase yields and apply other low-cost technologies [20].

Regarding the problems and causes that impede its development, which can affect the development of tourism together with agriculture and sustainability, the following was incorporated:

 Among the problems is the lack of knowledge on the part of the population about agro-tourism and the absence of places to carry out such activity adequately. The causes that limit the management of the activity may be the lack of preparation related to tourism activities, the scarcity of tourism development in the area, and the lack of tourism promotion.

With regard to the background of the research on agro-tourism in Cuba, the following content was included:

As background to the research, it can be said that since 1990 the Cuban state has deployed a political will to strongly develop the tourism sector in the country. In particular, the central region of Cuba has several historical sites typical of colonial cities, and its geographical enclave between the mountains and the coast is an outstanding destination which, together with sun and beach, nature, and cultural tourism, can be used for tourism development. In this context, there are state and privately owned farms, agricultural enterprises, and plantations with sufficient tourist attractions which are not being exploited and which can provide economic and social benefits for the people living in rural areas.

The following content was included for less-favored areas:

The Organisation for Economic Co-operation and Development defines the most disadvantaged areas as those where social, economic, and environmental problems are concentrated. Cuba's rural areas, as in other Latin American countries, are considered places with the most disadvantaged conditions from an economic, social, and environmental point of view. The insufficient availability of services associated with water, electricity, and sanitation coverage are problems that have accumulated from previous periods. The lowest salaries are located in the rural sector and the precariousness of work in agricultural activities means that the greatest poverty is located in the countryside (López-Chávez, 2020). The deterioration in the physical and constructive structure of housing and the poor state of communication routes with rural communities, together with the reduced transport coverage for the mobility of the population (Maceo, 2014; Sánchez, 2009; Peláez, 2016), create a real situation of isolation and marginality that worsens the living conditions of rural Cuba.

The World Tourism Organization's recommendations on tourism and rural development state that equitable distribution of tourism benefits can enhance job creation, protect natural resources and cultural heritage, promote social inclusion and empower local communities and traditionally disadvantaged groups such as women, youth, and indigenous peoples. Inclusive tourism can contribute to making rural territories more accessible to people and visitors from different generations and with varying needs of access, thus creating a better quality of life for all (World Tourism Organization [UNWTO], 2020).

In accordance with its guidelines, the content of the introduction was transferred to the conclusions, and the theoretical delimitation was placed after the introduction.

A new map was incorporated into the work to provide a better understanding of the context in which the study area is located.

New content was incorporated in Materials and Methods in order to clarify the current status and methodology used:

Hacienda Guachinango Constitutes a small agrotourist agricultural enterprise, where an old colonial structure that served as the home of the hacienda owners is located. The main agricultural activity is the cultivation and harvesting of sugar cane, small fruits, and occasional short-cycle crops for family consumption. Small and large livestock are raised to satisfy the needs of the family and to market the surplus.

In the surroundings of the Guachinango farm, there is a tourist center located in the rural village of Manaca Iznaga and two other farms with similar characteristics, where agro-tourism can be carried out.

There is a tourist train that runs through the Valle de los Ingenios and reaches the Hacienda Guachinango. This provides a reliable source of national and international tourists.

The Hacienda is located in a social context characterized by high levels of alcoholism, family conflicts, and crime. The main source of employment is agriculture, which does not meet the demand for labor. Low wages lead to a high level of precariousness in life. Poor families predominate, with incomes below the national average of 1.90 dollars a day [31]. The fundamental problems in the social sphere are the poor state of the roads, the lack of a water supply network, inefficient public transport services, the poor state of sports facilities, and deficient street lighting. All this demonstrates the existing difficulties in the public services offered by the state. It can be affirmed that Hacienda Guachinango is located in a disadvantaged area, which corresponds to the definition of the most disadvantaged areas by the Organisation for Economic Co-operation and Development, despite being located in the Valle de los Ingenios, which has been declared a UNESCO World Heritage Site since 1988.

The research was carried out at Hacienda Guachinango in the period from 2019 to 2021. The fieldwork made it possible to observe the environment within a radius of 4 kilometers and to extend the study to nearby communities and 2 farms that present a similar situation and where agro-tourism initiatives can be undertaken, as wages are low and farming families lead a precarious life that barely allows them to maintain low levels of agricultural production.

Three unstructured interviews were conducted with a purposive sample of 150 people, including 15 workers at Hacienda Guachinango, 23 local residents, 28 actors in the tourism sector at the municipal and provincial level, and 84 clients, in order to determine the strengths and weaknesses of the internal and external environment for the development of agro-tourism and its potential.

Two structured surveys were applied to a sample of 95 intentionally selected people. The first one to 15 workers of the farmhouse about the agro-tourism product and its future potential to maintain an adequate position in the market, and the second one to 80 national and international clients to measure the level of satisfaction achieved with the tourism experience.

A focus group discussion was carried out in two sessions. The first was held with the workers and the second session included tourism stakeholders from the provincial and municipal levels, in order to learn about and expand on ideas related to the attention and satisfaction of tourists and its link with the attention to agricultural production, its results and the participation of tourists in these activities.

Taking advantage of observation, fieldwork, interviews, surveys, and focus group results, a resource inventory was structured as a dynamic tool (Valseca-Martín, 2009; Blanco-López et al, 2015) in order to establish a diagnosis related to the potential of Hacienda Guachinango to develop agro-tourism, satisfy the preferences and expectations of tourists and make decisions in this regard.

 A marketing diagnosis was carried out in order to examine the tourism product, price, distribution, and communication (Arosa-Carrera, 2010; Ascencio-Hernández, 2016) at Hacienda Guachinango, as the main aspect of marketing planning. Based on the results, a mapping of the four fundamental marketing variables was drawn up to identify possibilities for improvement through duly planned strategic actions. A comprehensive diagnosis of the current situation was carried out, the main problems and latent potentialities, strengths and weaknesses, threats and opportunities were identified through the application of a brainstorming exercise carried out with the participation of the workers of the Hacienda and the actors of the tourism sector at the municipal and provincial level, all of which allowed the ideas to be organized in order of priority.

For the application of the unstructured interviews, the interview guide was used, differentiating the topics for the tourists from those discussed with the workers of the Hacienda and the actors of the sector at the municipal and provincial levels.

From the results of the unstructured interviews carried out with the selected sample, it was found that international and national tourists showed satisfaction with the agro-tourism product offered and referred mainly to the architectural and scenic value of the site, the natural and environmental benefits offered by the surroundings, as well as the possibilities of really getting to know the way of life in the Cuban countryside and understanding the real needs of the resident population.

The results of the surveys carried out with the workers revealed the material limitations related to the logistics of providing a good service to tourists. They recognized the need to improve the quality of services and complimentary offers, improve infrastructure, to fix access roads, and signposting to and from the Hacienda. The need to develop strategic alliances with nearby tourist centers, as well as with the hostels in the city of Trinidad, and to take advantage of the opportunities offered by the tourist train, was also raised. The need to adapt the offer to different market segments and to reconsider national tourism as a very important market was recognized.

The opinions of international and national tourists are positive and favorable. Aspects that need to be improved include the need for greater promotion and marketing of the site, better transport options to visit neighbouring communities, the improvement of access roads and signposting, as well as greater diversification of the offer.

The results of the focus group showed that there are criteria that coincide with the results of the surveys and interviews. The role of local governments and the need for strategic alliances between the public and private sectors emerged as a new element, as well as the need to encourage greater participation of communities and farmers in agro-tourism projects, which could improve the living conditions of families and community development in the area.

The inventory as a register made it possible to organize the set of attractions that form part of the agro-tourism product and to evaluate and rank the attractions. As a result, the different existing natural and cultural resources were evaluated and ranked, and their potential for attracting tourists was determined.

The marketing diagnosis allowed for a detailed understanding of the way in which Hacienda Guachinango is seen by its public. This is an essential aspect of marketing planning. As a result, a mapping of the four marketing variables was obtained in order to identify possibilities for improvement through future strategic actions. A comprehensive diagnosis of the current situation was made, and the main problems and potential opportunities, as well as strengths and weaknesses, threats, and opportunities, were identified through brainstorming. List reduction and weighted voting allowed the ideas to be organized in order of priority. The main problems affecting the marketing variables were identified and are summarised in table 2 and figure 8 of the paper.

New content was incorporated into the conclusions:

The analysis of the socio-economic context of Hacienda Guachinango revealed that in its surroundings there is a high level of alcoholism, social conflict, a high level of petty crime, and few employment opportunities with low wages. Except for electricity, the area does not have the rest of the basic services and all this describes a climate of precariousness and poverty in the peasant families, so it can be seen that the area has the characteristics of being one of the least favored, despite the fact that the Valle de los Ingenios was declared a World Heritage Site by UNESCO in 1988.

It is concluded that in the Hacienda Guachinango, the conditions required to pro-mote agrotourism are met as an activity that is born within the rural family capable of generating economic income that makes it possible to reduce the precariousness of life in the countryside and redefine the natural, historical, social, and productive values with the aim of increasing agricultural production and reducing migration to the cities. The applied methodology and the experiences derived from the research can be applied in similar rural contexts.

It was necessary to consult 29 other bibliographic documents, 20 of which were referred to and incorporated into the work, which reinforce and validate the information provided.

We reiterate our thanks and remain at your disposal.

PhD. Antonio Vázquez Pérez.

Reviewer 2 Report

This manuscript deals with an interesting topic and also addressed to aims and scope of this Journal.  Yet, we suggest some important improvements mainly on the literature review, methodology, conclusions, and reference sections.

Beginning with the literature review. Is needed a deeper literature revision, particularly on motivations and satisfaction. Also, the methodology should be better explained with the following points: survey development and data collection. Specifically, there’s a lack of information about the scales used on the questionnaire (based on what previous research works?) and how the study was conducted? (“intentionally selected samples of international and national tourists”. I don’t understand…). 

The results are not supported by literature review. How the results are connected with the literature review? Also, the conclusion part is weak because it's not clearly identified the effective theoretical implications to tourism literature development. Moreover, there are no study limitations presented in this manuscript. Study limitations are an important section to a better understanding of the results and possible generalizations.

Finally, the manuscript must be written in correct English and References are not accordingly to the format request. The authors provide different formats for the same type of references. 

In conclusion, this manuscript has some value, however, needs considerable (scientific) work.

Author Response

Good afternoon.
We are very grateful for your feedback and recommendations. This provided a theoretical improvement and enrichment of the work.
On the basis of the points made, the paper was restructured. Thus the theoretical delimitation (theoretical framework) was placed after the introduction and enriched by a broader literature review. This implied the consultation of 25 new bibliographical sources, of which 21 were cited. Thus, between the introduction and the theoretical delimitation, more than 85% of the literature referred to in the article was cited, 97 bibliographical documents in total.
In terms of methodology, the following content was incorporated, which provides more clarity on the methods used in the research:

  1. Materials and Methods

2.1. Study Area

Hacienda Guachinango is in the central west region of Valle de los Ingenios, declared a World Heritage Site by UNESCO in 1988, belonging to the municipality of Trinidad in the province of Sancti Spíritus, Cuba. It is surrounded by an agricultural area of 190 hectares of mostly typical tropical brown soils, with real possibilities to develop the traditional cultivation of sugar cane. It is characterised by tropical grasslands suitable for pastures and, to a lesser extent, typical alluvial soils, on which vegetables and fruit trees are grown. The Ay River maintains the water balance necessary to sustain the diversity of agricultural production throughout the year. Figure 1 shows the location of Hacienda Guachinango.

Figure 1. Location of Hacienda Guachinango.

Source: own elaboration from [86].

Hacienda Guachinango was built on a small hill 15.8 m high, and its heritage value is given as it is the only exponent of a linked dwelling with the rural architecture of the first half of the 19th century in the Valle de los Ingenios. It is a small agrotourist agricultural enterprise, where an old colonial structure that served as the house of the landowners is located. The main agricultural activity is the cultivation and harvesting of sugar cane, small fruits and occasional short-cycle crops for family consumption. Small and large livestock are raised to satisfy the needs of the family and to market the surplus.

In the vicinity of the Guachinango hacienda, there is a tourist centre located in the rural village of Manaca Iznaga and two other agricultural estates with similar characteristics, where agrotourism ventures can be carried out.

There is a tourist train that runs through the Valle de los Ingenios and reaches the Guachinango Farm. This provides a reliable source of national and international tourists.

Located at approximately 16 km from the city of Trinidad, the farm can be easily reached by bus or light vehicle through the highway from Trinidad to Manaca Iznaga and Condado, as well as by taking the tourist train that runs through the Valley of the Wits.

The Hacienda is in a social context characterized by high levels of alcoholism, family conflicts, and crime. The main source of employment is agriculture, which does not cover the demand for work. The low wages lead to high precariousness of life. Poor families with incomes below the country's average, which is USD 1.90 per day, predominate [87]. The fundamental problems in the social sphere are in the poor condition of the roads, lack of an aqueduct network, inefficient public transport service, poor condition of sports areas, and poor public lighting. All this demonstrates the existing difficulties in the public services offered by the state. This is why it can be affirmed that Hacienda Guachinango is located in a disadvantaged area, which corresponds to the definition of the most disadvantaged areas by the Organisation for Economic Cooperation and Development, despite the fact that it is located within the Valle de los Ingenios, which has been declared a UNESCO World Heritage Site since 1988.

2.2.        . Methodological Framework

The research was carried out at Hacienda Guachinango in the period from 2019 to 2021. The fieldwork made it possible to observe the environment within a radius of 4 kilometres and to extend the study to nearby communities and 2 farms that present a similar situation and where agrotourism initiatives can be undertaken, as wages are low and farming families lead a precarious life that barely allows them to maintain low levels of agricultural production.

In order to contribute to the achievement of the proposed research objective, an exploratory and descriptive study based on the mixed paradigm (linking qualitative and quantitative) was projected on the basis of a qualitative analysis and description of the current situation of Hacienda Guachinango, with the aim of identifying the potential to satisfy the demand for agrotourism [88].

The synthetic analytical method was applied and diagrams were drawn up and recorded in the research logbook. This required observation during fieldwork, as well as the application of interviews, surveys, the formation of focus group discussions and brain-storming with actors directly related to the tourism resource. Descriptive statistics were used to process the results obtained. Figure 2 shows the diagram of the applied research methodology.

Figure 2. Diagram of the research methodology.

Source: own elaboration based on [88].

To meet the stated objectives of the work, several data collection methods were designed and applied, including three semi-structured interviews, two surveys, a focus group that included two sessions and a brainstorming session, all of which were necessary to collect information that was used in the elaboration of the resource inventories, the diagnosis and to draw up the improvement plan for the enhancement of the agrotourism product at Hacienda Guachinango.

A non-probabilistic sample of 150 people was selected for the interviews, including: a group of 15 workers from Hacienda Guachinango and 28 actors from the tourism sector at the municipal and provincial levels; a group of 23 local residents; and a third group of 84 clients. The interviews were aimed at ascertaining the strengths and weaknesses existing in the internal and external environment for the development of agrotourism and its potential. The instrument used was a semi-structured interview guide. A Microsoft Excel spreadsheet was used for the statistical management of the information and its qualitative interpretation. 

Two structured opinion surveys were applied to a non-probabilistic sample of 95 people. In the first one, all the workers (15 employees) of the Hacienda were considered; in the second one, 80 national and international clients were considered to measure the level of satisfaction achieved with the tourist experience. The surveys were focused on measuring the agrotourism product and its quality, as well as the future potential to maintain an adequate position in the market and to enhance the value of the product. The instrument used was the survey questionnaire, for the workers 15 structured questions were designed and for the tourists 19 structured questions were prepared in Spanish, English and German. A Microsoft Excel spreadsheet was used for statistical data management and calculations.

Both the interviews and the surveys were not administered at a single point in time, especially those taken with the tourists. It was necessary to visit the Hacienda several times between 2019 and 2020. In this sense, it was possible to provide information for the collection and analysis of data aimed at strengthening the qualitative criteria of the diagnosis on the performance of agritourism at Hacienda Guachinango.

The selection of non-probabilistic sampling responded to the fulfilment of the objective proposed for the work, since the aim was to obtain the cases that were of interest for the research and that offer a great wealth for the collection and analysis of the necessary information on the agrotourism product at Hacienda Guachinango [88].

In order to strengthen the qualitative component of the work, a focus group discussion was held in two sessions, one in 2019 and the other in 2020. The first included farm workers, while the second session considered tourism stakeholders from the provincial and municipal levels, to learn and expand on ideas linked to the attention and satisfaction of tourists and their link with agricultural production, its results and the participation of tourists in these activities.

The brainstorming was carried out with the purpose of enriching the information on the tourist product and the possibilities of diversifying the offer, based on the tourist resources of the Hacienda and the surrounding environment. A carefully prepared questionnaire was prepared for this purpose. The questions were formulated in an open-ended way by means of a free-form discussion with all participants.    

By means of observation, fieldwork, interviews, surveys, focus groups and brainstorming, the results were triangulated, which allowed the structuring of a resource inventory as a dynamic tool [89; 90] in order to establish a diagnosis related to the potential of Hacienda Guachinango to develop agrotourism, satisfy the preferences and expectations of tourists and make decisions in this regard.

 A marketing diagnosis was carried out in order to examine the four variables such as tourism product, price, distribution and communication [91; 92] as the main aspects for marketing planning. Based on the results, a mapping of the four key variables was developed in order to identify possibilities for improvement through properly planned strategic actions. A comprehensive diagnosis of the current situation was carried out, identifying the main problems and latent potentialities, strengths and weaknesses, threats and opportunities. All this was done with the participation of the workers of the Hacienda and the actors of the tourism sector at the municipal and provincial level, which made it possible to organize the ideas in order of priority.

2.3. Data and Data Sources

Based on the diagnosis, a matrix of weaknesses and strengths (internal environment variables) and threats and opportunities (external environment variables) was created, with each assigned to the registered items in each category and the value of its impact on the tourist resource.

The classification of the strength items was as follows: high impact 4 points, moderate impact 3 points; for weaknesses: moderate impact 2 points, low impact 1 point; in the threat items: very high impact 4 points, high impact 3 points; and for the opportunities: regular impact 2 points, little impact 1 point.

Subsequently, each of the items corresponding to the internal environment was weighted with a relative value, the sum of which was equal to one (Σ=1). The same procedure was followed with the items referring to the external environment.

The sum of the values obtained by multiplying the value of the impact assigned to each item of the internal environment by its weighting revealed the value of the competitiveness of the resource on a scale of 0 to 4 points. The location of both scores in the corresponding quadrant shed light on the strategy to follow.

The factors that have favoured the development of agrotourism are not only on the supply side but also on the demand side [75]. Therefore, an analysis of the market was carried out according to supply and demand, as well as studies on image and satisfaction, to redesign the Hacienda Guachinango tourist product, as well as a study of the situation presented by the marketing variables, including product, price, distribution, and communication.

To analyse the behaviour of the demand, the logical historical method was applied, accessing the statistical data registered in the Territorial Office of Tourism. The data quantified for this purpose were considered according to the system of indicators established for their records, such as inclusion of the technique of inventorying tourist resources and attractions, with their justification; a diagnosis of the product, price, distribution, and communication; market segmentation to determine the main segments of agrotourism and main markets that visit Hacienda Guachinango; the analysis of the surveys to determining to what extent the product constituted a special attraction during and after the COVID-19 pandemic; definition of the contribution of tourism to alleviate poverty and its contribution to depressed areas; and the application of a diagnostic technique that favours defining the strategy to follow, based on the variables of tourism in the territory.

For this purpose, two structured surveys were applied to a non-probabilistic sample of 95 people, including 80 international and national tourists and 15 employees of the Hacienda during the years 2019 and 2020. The first was focused on obtaining information on tourist preferences and the second to determine the degree of satisfaction of visitors who chose the Hacienda Guachinango tourism product.

For this, two surveys were applied to an intentionally selected sample of international and national tourists during the years 2021 and the first quarter of the year 2022. The first focused on obtaining quantitative information on the preferences of tourists and the second on determining the degree of satisfaction of visitors who chose the Hacienda Guachinango tourism product.

The classification of the items was carried out to determine the motivational factors and satisfaction of the tourists, with the application of an ordinal scale where the satisfied and unsatisfied demands on the site were determined. The following scale was employed: very satisfied 5, satisfied 4, neutral 3, dissatisfied 2, totally dissatisfied 1, with a margin of error = 4%, level of confidence = 95%, and origin of the issuing market (foreign and national) = 50%.

As an inclusion criterion, all tourists present at the time of the survey were considered, provided they gave their consent and after having enjoyed the products offered at Hacienda Guachinango. As an exclusion criterion, foreign and national tourists who did not give their consent to participate were not included in the study as well as those newcomers who had not enjoyed the product offer and who, for obvious reasons, were unaware of it. For the application of the unstructured interviews, the interview guide was used, differentiating the topics for the tourists from those discussed with the workers of the Hacienda and the actors of the sector at the municipal and provincial levels.

A correlation was established between the content of the results and the literature consulted, in order to establish a theoretical relationship between the results and the literature cited. 
The conclusions section was enriched and the limitations of the study were included, which provides a better idea of the results and possible generalisations, as follows:

Conclusions

The fulfilment of the objectives set for the work made it possible to characterise the agrotourist product of the Guachinango Hacienda in the Valle de los Ingenios in the municipality of Trinidad, which made it possible to understand its significance, relevance and perspectives for the development of the agro-tourist modality in the context of rural tourism, by integrating into it the historical, cultural and patrimonial values treasured for more than four centuries of existence. To this must be added the values of the daily work of the small family dedicated to agricultural production, with the possibility of the tourist being able to share unprecedented experiences in an environment of family production, of cultural exchange and generator of positive synergies for the tourist and agricultural production.

The development of the objective of the work revealed that the agro-tourist enterprise of Hacienda Guachinango has a staff of 15 workers, of which 2 are directly responsible for attending to tourism. The rest are dedicated to agricultural production and the care of large and small livestock. The income derived from the tourist activity is added to the general account of the farm, which improves the income of the staff, achieves greater equity and improves the living conditions of the people. Another advantage is the reinforcement of the amount of investments of the farm in order to achieve higher and sustainable production. In this way, the theoretical study of the agrotourism modality at Hacienda Guachinango allowed us to know that the tourism enterprise does not affect the social destination of agricultural production, on the contrary, it strengthens it in correspondence with the criteria provided by [5; 7; 8; 9; 10; 11; 12; 14] in contrast to what has been proposed by the authors [15; 16]. From this stems the relevance of the theoretical study of the agrotourism modality in Cuba, as a new phenomenon that stems from the emerging need to develop tourism as an activity that guarantees the economic progress of the country.

The analysis of the socio-economic context of Hacienda Guachinango revealed that there is a high level of alcoholism, social conflict, a high level of petty crime and few employment opportunities with low salaries. Except for electricity, the area does not have the rest of the basic services and all of this describes a climate of precariousness and poverty in the peasant families, so it can be seen that the area has the characteristics of being one of the least favoured, in correspondence with the World Tourism Organisation [46], de-spite the fact that the Valle de los Ingenios was declared a World Heritage Site by UNESCO in 1988.It is concluded that in the Hacienda Guachinango, the conditions required to pro-mote agrotourism are met as an activity that is born within the rural family capable of generating economic income that makes it possible to reduce the precariousness of life in the countryside and redefine the natural, historical, social, and productive values with the aim of increasing agricultural production and reducing migration to the cities. The applied methodology and the experiences derived from the research can be applied in similar rural contexts.

The work revealed that there is a wide range of resources and attractions that could be used to establish a post-COVID-19 agrotourism reactivation model for rural areas within the framework of sustainability. It was discovered that in the area surrounding Hacienda Guachinango there are two other farms with real possibilities for agrotourism business. The results of the research can serve as a theoretical starting point for further study of the modality and be applied in the two farms mentioned above, as well as in other similar environments in the country and Latin America, where there are many peasant families who daily face the precariousness and high cost of living that they have to pay for the fact of dedicating themselves to small-scale agricultural production in a rural area. All of this in spite of having natural, historical and cultural riches similar to those found in Hacienda Guachinango. 

The results of the surveys carried out with foreign and national tourists showed that Hacienda Guachinango has a high motivational attraction for the enjoyment of the tourist products on offer. Resources that should be adequately exploited in accordance with what is stated by [34].

The research revealed the competitiveness of Hacienda Guachinango and the scope to develop the tourism modality of agrotourism, given its current capacity and potential to develop competitive advantages over its competitors by obtaining a prominent position in terms of the quality and variety of tourism products that are unique in their environment.

The systemic analysis of the different applied research methods identified the real perspectives of agrotourism in the Hacienda Guachinango as a dynamic element of agri-cultural production to improve the socioeconomic conditions of small peasant families that can benefit collaterally with the additional income generated by the tourist modality in less favoured areas to benefit the living and working conditions of the peasant.

However, the theoretical study of agrotourism is not free of limitations. Among them is the lack of experience with this recently emerging tourism modality, which does not have sufficient theoretical studies that address the subject in a comprehensive manner.

The study of agrotourism is inadequately appreciated by the tourism sector as a marginal problem of a local nature, which does not require priority attention for research and development, as well as the promotion of the tourism value of the resources that exist in the rural environment.

Another limitation is related to the lack of investment by the State to improve roads, transport services to and from rural communities, the poor coverage of Internet communication, the low quality and reliability of electricity services, the scarce access to bank credit at low interest rates that would allow facilities to be equipped for tourism, and the lack of commercial networks to guarantee the inputs required for visitor services, are all limitations that affect the generalisation of the agrotourism modality, in accordance with what has been stated by [33].

Particularly in Cuba, the poor working relationship between the public and private sectors is another limitation to the study of agrotourism. In the country, the opening of businesses to the private sector is a recent phenomenon that has not yet achieved the necessary experience to provide a wide opening to private enterprises, which is reflected in weaknesses in promoting public policies aimed at benefiting private business.  

From the internal point of view, the lack of preparation of the people directly involved in the management of the tourism activity, a situation that constitutes a priority and that should be resolved through alliances with the tourism sector aimed at the training of these people.

The lack of understanding of the importance of national tourism compared to foreign visitors, which can be seen in the results of the survey on satisfaction with the enjoyment of tourism products, where foreign tourists have a much higher level of demands and requirements for the consumption of agrotourism products compared to national tourists. It was found that the actors in the tourism activity do not treat national tourists in the same way as foreign tourists.

The references conformed to the requested format. 
We reiterate our thanks for the attention given to our work.

PhD. Antonio Vázquez Pérez

Round 2

Reviewer 1 Report

The authors have well revised the paper, providing deeper research background, detailed explanation of the research methodology and extended conclusion. 

There are some minor changed are suggested to be made before publishing:

- consider shortening the list of numerous references in a single sentence (e.g. [5,8,19,20,23,24,25,26] or [35,36,37,38,39,40,41]). There is no need to mention everyone who has done research on the topic. Ideally is to keep just a few major one and give a reference describing the actual contribution. This is especially important in the case of prior research on Cuba.

- ideally to move all references from the Results to Discussion section (e.g. in line with what has been suggested by the 470 authors [5; 6; 8; 19; 20]. OR as suggested by 479 [69; 70] etc.). This will make richer Discussion and make authors' findings clearer.

- Use high resolution figures; revise them on spelling (e.g. 4,7 change to 4.7 in Figure 9).

- Revise formatting the References section (e.g. Barbieri, C; Xu, S; Gil-Arroyo, S.R. Rich. Agritourism, farm visit, or...? A branding assessment for recreation on farms. J. Travel 839 Res. 2015. 55(8), P. 1094-1108. Terms and conditions | Taylor & Francis Online (tandfonline.com). 840 https://doi.org/10.1177/0047287515605930. - why mention "Terms and conditions | Taylor & Francis Online (tandfonline.com)"?) 

Additionally, please consider strengthening your statement about less favored areas. To my mind, Hacienda Guachinango de Trinidad can hardly be regarded as less favored area, especially for rural tourism. It is located next to the national highway, having roads, railroad, featuring cultural and historical monuments. Rural areas do have lower density of roads and public infrastructure, smaller wages and no Internet. But this is normal throughout the world, it make countryside different from cities.

Author Response

Good afternoon.
Your suggestions and recommendations have been well received. We are very grateful for your work and the help you have given us.
With regard to your remarks and recommendations, we would like to inform you of the following:
The list of numerous references was shortened to one sentence.
The references that were in the results section were moved to the discussion section, which made the results clearer and at the same time enriched the discussion of the paper.
The resolution of figure 9 was improved.
The format of the references section was revised.
Comments
After a thorough analysis and discussion among the authors through the study of the literature and the fieldwork carried out, the following problems were found to be present in Hacienda Guachinango, which are coherent with the definition and indicators that identify disadvantaged areas: 
Low level in living conditions and social coexistence in the area, low cultural level, high unemployment rate, and presence of alcoholism in young people. Poor housing infrastructure. The poor state of roads, deficient public transport service. Low investment in human capital is a key to empowering people and communities. Little social innovation-seeking new solutions to societal issues that are successful and have a positive impact on individuals and communities, especially vulnerable groups. The little social will to undertake the collective effort. Little presence of the state and the private sector to deploy sustainable economic and community development policies in the area.
The specific analysis of Hacienda Guachinango and the socio-economic consideration of its surroundings led to the conclusion that the Hacienda is located in a disadvantaged area, which was corroborated in a study presented by the Centro de Estudios de la Economía Cubana of the University of Havana [43]. 
It is precisely the revival of the tourist offer, the quality of the service, the use of the natural potential, and the opening of agro-tourism in the two nearby farms, which can provide a leap in the quality of life of the inhabitants of the area, reduce unemployment, achieve greater equity in the distribution of income, based on the maintenance, improvement, and expansion of the agricultural activities that are carried out. For this reason, the work states that agro-tourism can be considered a first-order solution for less-favored areas.

PhD. Antonio Vázquez Pérez

REVISED AND ELABORATED
13 July 2022 18:30 h
